# Topological Anomaly Quantification for Semi-supervised Graph Anomaly Detection

**Ting Guo[1], Yangrui Fan[2], Caixia Cui[2], Jiye Liang[3*], Jiao Zhao[3], Da Wang[3]**
[1] School of Computer Science and Technology, North University of China,
Taiyuan, 030051, Shanxi, China
[2] College of Computer Science and Technology, Taiyuan Normal University,
Taiyuan, 030619, Shanxi, China
[3] Key Laboratory of Computational Intelligence and Chinese Information Processing of Ministry of Education, School of Computer and Information Technology, Shanxi University,
Taiyuan, 030006, Shanxi, China
E-mails: `gting@nuc.edu.cn` (Ting Guo);
`ljy@sxu.edu.cn` (Jiye Liang, corresponding author)

## Abstract

Semi-supervised graph anomaly detection identifies nodes deviating from normal patterns using a limited set of labeled nodes. This paper specifically addresses the challenging scenario where only normal node labels are available. To address the challenge of anomaly scarcity in real-world graphs, generative-based methods synthesize anomalies by linear/non-linear interpolation or random noise perturbation. However, these methods lack a quantitative assessment of anomalies, hindering the reliability of the generated ones. To overcome this limitation, we propose a generative graph anomaly detection model based on topological anomaly quantification (TAQ-GAD). First, we design a topological anomaly quantification module (TAQ), which quantifies node abnormality through two topological metrics: The node boundary score (NBS) quantifies the boundaryness of a node by evaluating its connectivity to labeled normal neighbors. The node isolation score (NIS) assesses the structural isolation of a node by evaluating its connection strength to other nodes within the same category. This anomaly measurement module dynamically screens nodes with high anomaly scores as pseudo-anomaly nodes. Subsequently, the topological anomaly enhancement (TAE) module generates virtual anomaly center nodes and constructs their topological relationships with other nodes. Finally, the method integrates normal and pseudo-anomaly nodes on the enhanced graph for model training. Extensive experiments on benchmark datasets demonstrate TAQ-GAD's superiority over state-of-the-art methods and effectively improve anomaly detection performance.

## 1 Introduction

Graph anomaly detection (GAD) is a technique that aims to identify abnormal nodes in graphs that deviate from normal behavioral patterns Qiao et al. (2025); Tang et al. (2023); Liang et al. (2024). This technology is widely applied in critical domains such as financial fraud detection Liu et al. (2023), network intrusion prediction Wang et al. (2023a), and failure prediction Li et al. (2022), playing an important role in ensuring data security and improving system reliability. Despite significant progress in unsupervised GAD methods Ma et al. (2025); Li et al. (2025); Dong et al. (2025); Lin et al. (2024), their lack of labeled guidance confines them to intrinsic graph information, thus introducing fundamental ambiguity in identifying true semantic anomalies. In contrast, the semi-supervised GAD framework leverages limited labeled data (including both normal and anomalous nodes, or only labeled normal nodes) offers a more promising research direction. In vanilla semi-supervised GAD Ma et al. (2024); Chen et al. (2024); Liu et al. (2022a); Huang et al. (2022a); Gao et al. (2023); Shi et al. (2022), the availability of both labeled normal and anomalous node frames the problem as a binary classification task. However, obtaining reliable anomalous node labels is often impractical due to their rarity and the high cost of annotation. **The semi-supervised setting**

**with only normal labeled nodes** presents a more challenging yet realistic scenario, better reflecting real-world constraints where labeled anomalies are scarce or unavailable Qiao et al. (2024). The key advantage of this paradigm lies in its ability to leverage easily attainable normal nodes as reliable anchors, shifting the objective from identifying global outliers to detecting deviations from a learned normal profile. This transition not only markedly reduces the false positive rate and enhances robustness against sophisticated camouflaged attacks, but also aligns well with the practical constraints of real world labeling costs. Therefore, this work focuses on this more challenging and practical setting, aiming to explore how limited normal prior knowledge can be utilized to achieve more reliable anomaly detection. In order to alleviate the problem of scarce labels, generative-based methods learn latent distribution features of normal data and synthesize diverse anomaly samples to augment training data, thereby effectively improving model detection performance.

Existing generative-based methods mainly fall into two categories: feature interpolation methods and noise perturbation methods Qiao et al. (2024). Feature interpolation methods enhance distribution modeling and generalization by generating synthetic samples through linear/non-linear interpolation between node features Fernández et al. (2018); Zhou et al. (2024). However, the generated samples are often overly smooth, not only lacking the complexity of real anomalies and thus failing to capture boundary cases effectively, but also introducing additional computational overhead. Noise perturbation methods construct anomaly samples by adding random noise to the labeled normal nodes Cai et al. (2025); Chen et al. (2024); Liu et al. (2022a); Qiao et al. (2024). The core assumption underlying these methods is that anomalous samples exhibit significant deviations from the normal data distribution in either feature or structural space. Consequently, they typically simulate anomalous patterns by artificially injecting random perturbations into the feature space or graph structure of normal nodes. Despite their contributions, both strategies face inherent limitations in ensuring the quality of generated anomalies. A fundamental issue lies in the lack of a quantitative mechanism for assessing node anomaly degrees. The injected perturbations are largely stochastic and unguided, leading to generated pseudo-anomalous nodes that suffer from low representativeness and reliability. Consequently, these synthetic samples ultimately fail to simulate the complex and meaningful anomalous patterns observed in real-world scenarios.

Therefore, this paper proposes a topology-based pseudo-anomaly generation approach that selects high-quality pseudo-anomalies by deeply mining graph structural information. Specifically, we design two metrics: node boundary score (NBS) and node isolation score (NIS). NBS quantifies the boundaryness of a node by evaluating its connectivity to labeled normal neighbors. The connection density between anomalous nodes and normal nodes is substantially sparser than that observed among normal nodes themselves. The node isolation score (NIS) quantifies a node's isolation by measuring its connection strength to other nodes within the same category. We harness these combined metrics to select nodes topologically akin to anomalies from labeled normal data, leveraging them as pseudo-anomalies to improve the model's discriminative capability. Additionally, we further adopt a topological anomaly enhancement (TAE) module that dynamically generates virtual anomaly center points based on node prediction confidence and constructs their topological relationships with other nodes. Finally, the method integrates normal nodes and pseudo-anomaly nodes on the enhanced graph structure for joint training to achieve more accurate anomaly detection. The main contributions of this paper are as follows:

- We propose TAQ-GAD, a generative GAD model based on topological anomaly quantification, effectively addressing the low quality of generative nodes.

- We propose the node boundary score (NBS) and the node isolation score (NIS), two topological metrics that characterize a node's anomaly degree from the perspectives of boundary proximity and structural isolation, respectively.

- TAQ-GAD incorporates a TAE module that dynamically generates virtual anomaly center nodes and establishes their topological relationships with other nodes, significantly enhancing model anomaly identification performance.

- Extensive experiments on several datasets demonstrate that TAQ-GAD outperforms state-of-the-art methods, confirming its effectiveness for GAD.

## 2 RELATED WORK

### 2.1 GRAPH ANOMALY DETECTION

**Unsupervised GAD.** Unsupervised GAD methods detect anomalies by modeling the intrinsic structure of graphs without labeled data Qiao & Pang (2023). Reconstruction-based models are widely used—e.g., DOMINANT Ding et al. (2019) employs graph autoencoders, while AnomalyDAE Fan et al. (2020) enhances reconstruction with attention. Contrastive methods are also prominent: CoLA Liu et al. (2022b) compares nodes with local substructures, and GCCAD Chen et al. (2023) and ANEMONE Jin et al. (2021) learn from positive-negative pairs.

**Semi-supervised GAD.** Semi-supervised GAD leverages a limited amount of labeled data along with unlabeled nodes to improve detection performance. Existing methods can be categorized into two main paradigms: *Semi-supervised GAD with labeled normals and anomalies*, and *semi-supervised GAD with labeled normals only*. In the former category, methods exploit anomalies at both the feature and structural levels Ma et al. (2024); Liu et al. (2022a); Zhou et al. (2024); Tang et al. (2022); Huang et al. (2022a); Gao et al. (2023). On the feature side, CGenGA Ma et al. (2024) establishes design principles for denoising networks tailored to anomaly generation; gADAM Zhou et al. (2024) augments training data via adaptive interpolation in the embedding space; and BWGNN Tang et al. (2022) employs spectral filters to capture high-frequency anomaly signals. Structurally, AO-GNN Huang et al. (2022a) incorporates an AUC-oriented reinforcement learning module to optimize edge pruning policies; BHetero-GHRN Gao et al. (2023) trims inter-class edges by emphasizing high-frequency graph components to handle heterogeneity. In the latter category, where only normal labels are available, GGAD Qiao et al. (2024) proposes a generative framework that constructs pseudo-anomaly nodes to provide effective negative samples for training.

### 2.2 GENERATIVE GAD

To address the anomaly scarcity problem, generative-based methods synthesize pseudo-anomaly nodes to augment training sets. Based on synthesis strategies, existing methods are categorized into feature interpolation and noise perturbation approaches Qiao et al. (2025). **Feature interpolation methods** generate synthetic nodes by mixing features from normal instances Zhao et al. (2021); Wu et al. (2022). For example, GraphENS Park et al. (2022) introduced adaptive interpolation rates, and AuGAN Zhou et al. (2023) uses feature interpolation to generate synthetic anomaly nodes, augmenting training data for improved detection performance. **Noise perturbation methods** generate pseudo-anomaly nodes by injecting noise Chen et al. (2020). Representation permutation approaches include: DAGAD Liu et al. (2022a) employs the permutation and concatenates on the representation learned from a limited number of labeled instances to generate the anomalous sample, thereby enriching the knowledge of anomalies captured in the training set, GGAD Qiao et al. (2024) aims to generate outlier nodes that assimilate anomaly nodes in both local structure and node representations by leveraging the two priors of anomaly nodes, including asymmetric local affinity and egocentric closeness, to impose constraints the representations.

## 3 PRELIMINARIES

**Semi-supervised GAD.** We focus on semi-supervised GAD on graphs. Given an attributed graph $\mathcal{G} = (\mathcal{V}, \mathbf{A}, \mathbf{X})$ with $N$ nodes, where $\mathbf{A} \in \mathbb{R}^{N \times N}$ is a non-negative adjacency matrix and $\mathbf{X} \in \mathbb{R}^{N \times d}$ represents the node feature matrix. In the semi-supervised setting, we have access to a limited set of labeled normal nodes $\mathcal{V}_l \subset \mathcal{V}$, while the remaining nodes $\mathcal{V}_u = \mathcal{V} \setminus \mathcal{V}_l$ remain unlabeled. The goal of semi-supervised GAD is to learn an anomaly scoring function $f : \mathcal{V} \mapsto \mathbb{R}$ such that for any normal node $v$ and abnormal node $v'$, we have $f(v) < f(v')$, thereby effectively distinguishing normal nodes from abnormal ones. Following semi-supervised GAD settings, our method leverages normal labels and graph topology to identify high-confidence pseudo-anomalies.

**Graph neural networks.** We employ graph neural networks (GNNs) to learn node representations by aggregating structural and attribute information Guo et al. (2024); Liang et al. (2023). The layer-wise propagation is formalized as:

$$\mathbf{H}^{(\ell)} = \text{GNN}\left(\mathbf{A}, \mathbf{H}^{(\ell-1)}, \mathbf{W}^{(\ell)}\right), \tag{1}$$

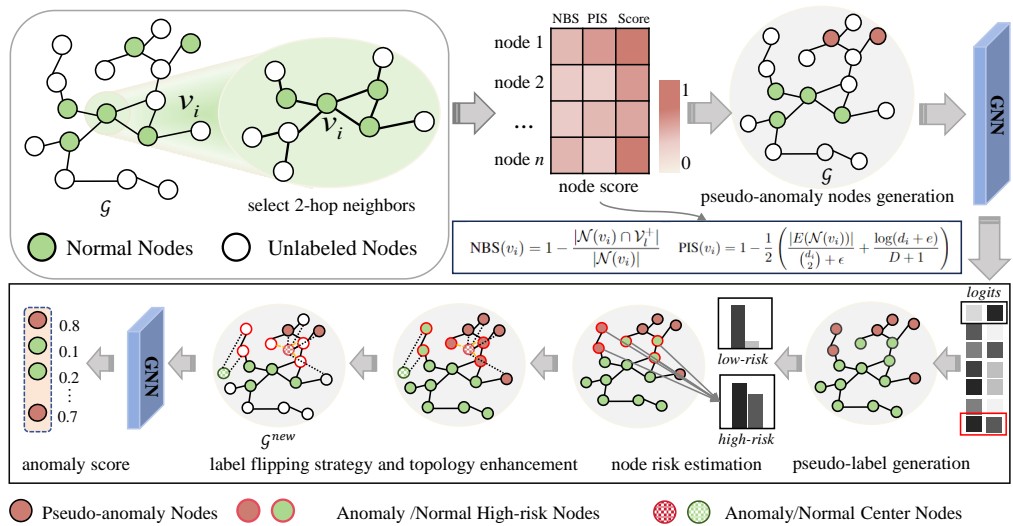

Figure 1: The overall framework of TAQ-GAD.

where $\mathbf{H}^{(0)} = \mathbf{X}$, and $\mathbf{H}^{(\ell)} \in \mathbb{R}^{N \times h^{(\ell)}}$ denotes node representations at layer $\ell$. The final representations $\mathbf{H}^{(L)} = \{\mathbf{h}_1, \ldots, \mathbf{h}_N\}$ with $\mathbf{h}_i \in \mathbb{R}^d$ are used for downstream tasks.

# 4 OUR METHOD

In this section, we first explore the correlation between graph topological structure information and node anomaly properties. We introduce the topological anomaly quantification module (TAQ). The TAQ module precisely quantifies the degree of node anomaly using NBS and NIS metrics. Then, the topological anomaly enhancement module (TAE) is designed to generate virtual anomaly center nodes and construct their topological relationships with other nodes. Finally, we theoretically analyze the rationality and the effectiveness of the proposed anomaly measurement metrics. Figure 1 illustrates the overall framework of TAQ-GAD.

## 4.1 TOPOLOGICAL ANOMALY QUANTIFICATION MODULE

In this section, we examine the relationship between graph topological structure and node anomaly properties. We introduce the TAQ module, which provides a precise quantification of node anomaly levels through the NBS and NIS metrics. Subsequently, the TAE module is designed to generate virtual anomaly centers and construct their topological relationships with other nodes. We further present a theoretical analysis demonstrating the validity of the proposed anomaly metrics and offer insights into the effectiveness of the TAQ-GAD. The overall architecture is illustrated in Figure 1.

Let $\mathcal{V}_l$ represent the labeled normal node set. For node $v_i \in \mathcal{V}_l$, we define the distance path$(v_i, v_j)$ as the shortest path length between nodes $v_i$ and $v_j$ in graph $\mathcal{G}$. We also construct the $K$-hop neighborhood of $v_i$ as: $\mathcal{N}(v_i) = \{v_j \in \mathcal{V} \mid \text{path}(v_i, v_j) \leq K\}$.

**Definition 3.1.** (Node boundary score, NBS). The node boundary score (NBS) of a node $v_i$ quantifies its proximity to the boundary of the normal region by measuring the proportion of labeled normal nodes among its neighbors:

$$\text{NBS}(v_i) = 1 - \frac{|\mathcal{N}(v_i) \cap \mathcal{V}_l|}{|\mathcal{N}(v_i)|}, \quad (2)$$

where $|\cdot|$ denotes the number of nodes in a set. A higher NBS value indicates that the node has fewer connections to the labeled normal class and is therefore more likely to reside near the decision boundary, suggesting a higher degree of anomalousness. This is illustrated in Figure 2, where node $v_i$ exhibits a higher NBS value than $v_j$, consistent with its sparser connections to normal nodes.

**Definition 3.2.** (Node isolation score, NIS). The node isolation score (NIS) of a node $v_i$ quantifies its structural isolation within its own category by measuring the mean shortest-path distance to other

nodes sharing the same category label:

$$\text{NIS}(v_i) = \frac{1}{|\mathcal{N}_s(v_i)|} \sum_{v_j \in \mathcal{N}_s(v_i)} \text{path}(v_i, v_j), \tag{3}$$

where $\mathcal{N}_s(v_i) = \{v_j \in \mathcal{N}(v_i) \mid y_{v_j} = y_{v_i}\}$ denotes the set of nodes within the $K$-hop neighborhood of $v_i$ that belong to the same category as $v_i$. A higher NIS value indicates that the node is more topologically isolated from other nodes in its category. As illustrated in Figure 2, node $v_i$ exhibits a higher NIS value than $v_j$, reflecting its weaker connections to other nodes of the same category, while $v_j$ maintains stronger topological associations with its peers. This suggests that $v_i$ has a higher likelihood of being an anomaly compared to $v_j$.

In the absence of genuine anomaly labels, we introduce the proxy isolation score (PIS), a topology-based metric designed to quantify node isolation. The PIS score is defined as:

$$\text{PIS}(v_i) = 1 - \frac{1}{2} \left( \frac{|E(\mathcal{N}(v_i))|}{\binom{d_i}{2} + \epsilon} + \frac{\log(d_i + e)}{D + 1} \right), \tag{4}$$

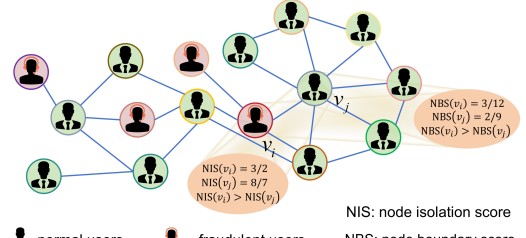

where $d_i$ denotes the degree of node $v_i$, $|E(\mathcal{N}(v_i))|$ is the number of edges among its neighbors, $D$ is the maximum degree in the graph, $\epsilon > 0$ is a small constant to avoid division by zero, and $e$ is the Euler's number. This formulation integrates two complementary as-

Figure 2: NBS and NIS metrics for quantifying the degree of node anomaly. The $\mathcal{N}(v_i)$ is defined as the 2-hop neighborhood of node $v_i$.

pects of topological sparsity: the first term captures the local clustering density, while the second term penalizes high-degree nodes to highlight structural sparsity. A higher PIS value indicates that the node has fewer neighbors and those neighbors are poorly interconnected, reflecting the "few and scattered" structural pattern characteristic of anomalous nodes. Although PIS is not mathematically equivalent to the supervised NIS metric, both capture the same underlying topological principle: anomalous nodes tend to be structurally isolated. Thus, in settings where anomaly labels are unavailable, PIS serves as a robust and practical surrogate. Its ability to leverage purely local structural information makes it especially suitable for large-scale graph-based anomaly detection.

Based on the $\text{Score}(v_i)$, we rank all labeled nodes and select the top $\tau$-proportion of highest-scoring nodes as pseudo-anomaly nodes for model training, where $\tau \in (0, 1)$ is a predefined ratio parameter.

$$\text{Score}(v_i) = \lambda_1 \text{NBS}(v_i) + \lambda_2 \text{PIS}(v_i), \tag{5}$$

where $\lambda_1$ and $\lambda_2$ are hyperparameters. After obtaining pseudo-anomalous nodes, we train the GNN model by integrating the remaining normal nodes and unlabeled nodes, subsequently generating prediction probability vectors for all nodes. NBS and NIS indicators are theoretically grounded in the structural characteristics of anomalous nodes in semi-supervised networks. We establish two key theoretical results that justify their effectiveness.

**Theorem 1 (NBS separability):** Under network homophily and labeling bias assumptions, anomalous nodes exhibit systematically higher NBS values than normal nodes, *i.e.*, $\mathbb{E}[\text{NBS}(v_i) \mid A_i] > \mathbb{E}[\text{NBS}(v_i) \mid N_i]$. Anomalous nodes at the network periphery have fewer labeled neighbors, so a higher proportion of their neighbors are unlabeled.

**Theorem 2 (PIS anomaly indication):** A high value of PIS (representing extreme topological isolation) significantly increases the posterior probability of a node being anomalous, *i.e.*, $\mathbb{P}(A_i \mid \text{PIS}(v_i) = 1) > \mathbb{P}(A_i)$. This phenomenon arises from the topological isolation of anomalous nodes, which are more likely to lack direct connections to labeled normal nodes.

These theoretical insights demonstrate that NBS captures the boundary characteristics of nodes (through neighborhood labeling density), while PIS captures their topological isolation (through complete disconnection from normal label support). Together, they provide a principled approach to identify pseudo-anomalous candidates from labeled normal nodes. The formal proofs, including detailed mathematical derivations and validity conditions, are provided in Appendix A.

## 4.2 Topological Anomaly Enhancement Module

The TAE module refines graph structure by strengthening connectivity between anomalous nodes and anomaly centroids, leading to improved anomaly detection performance. It operates in two stages: first, it estimates risk coefficients for nodes to derive pseudo-labels; then, it generates pseudo-class centroids based on the refined pseudo-labels and constructs topological relationships between the centroids and other nodes. By augmenting the graph with pseudo-labels and pseudo-anomaly centroids, the TAE module reinforces the topological connections of anomalous nodes, thereby enriching their informational context for more discernible detection.

For **node risk estimation**, let $P_{v_i} = [p_{v_i}^{(0)}, p_{v_i}^{(1)}]$ denote the predicted probability distribution of node $v_i$, with $p_{v_i}^{(0)}$ and $p_{v_i}^{(1)}$ representing the probabilities of belonging to the normal and anomalous classes, respectively. Based on this probability, we derive the initial pseudo-labels for nodes. The prediction uncertainty is quantified as: $u(v_i) = 1 - \max_{c \in \{0,1\}} p_{v_i}^{(c)}$, where $c$ indexes the class labels. This metric captures the model's confidence in its prediction for node $v_i$, with higher values indicating greater uncertainty. To mitigate selection bias arising from inter-class uncertainty disparities, we compute a risk score relative to the class-wise average uncertainty:

$$r(v_i) = \max\left(0, u(v_i) - \bar{u}_{\hat{y}_i}\right) \times w_{\hat{y}_i}, \tag{6}$$

where $\hat{y}_i$ is the predicted class of $v_i$, $\bar{u}_{\hat{y}_i}$ denotes the average uncertainty across nodes predicted as class $\hat{y}_i$, and $w_{\hat{y}_i}$ is a class weight based on inverse frequency calculated as the ratio of the smallest class size to the current class size. $w_{\hat{y}_i}$ tends to assign higher importance to the minority (anomalous) class. A high risk score $r(v_i)$ indicates that the node's uncertainty substantially exceeds its class average, flagging it as potentially misclassified or ambiguous.

Subsequently, we leverage the graph's topological structure to compute the class distribution within each node's neighborhood as a posterior probability:

$$p_{v_i}^{\text{post}(c)} = \frac{|\{v_j \in \mathcal{N}(v_i) : \hat{y}_j = c\}|}{|\mathcal{N}(v_i)|}, \quad c \in \{0, 1\}, \tag{7}$$

For high-risk nodes (where $r(v_i) > 0$), we implement a **label flipping strategy**: if the node's neighbors show high confidence in the opposite class (i.e., $p_{v_i}^{\text{post}(1-\hat{y}_i)} > p_{v_i}^{\text{post}(\hat{y}_i)}$), we flip the node's pseudo-label from its current prediction $\hat{y}_i$ to the opposite class $1 - \hat{y}_i$. This approach is based on the topological consistency principle. When neighbor information strongly supports the opposite class, it indicates the original prediction may be incorrect and should be corrected through label flipping. By applying a label flipping strategy, we generated the refined pseudo-labels.

Based on the current labeled nodes, we compute class centroids for each category. The connection probability between node $v_i$ and centroid $c$ is defined as:

$$P(v_i, v_c^{\text{virtual}}) = r(v_i) \cdot p_{v_i}^{\text{post}(c)} \cdot (\mathbf{1} - \mathbb{I}[\hat{y}_i = c]). \tag{8}$$

Through probabilistic sampling, we generate virtual edges between nodes and centroids, constructing an augmented graph $\mathcal{G}^{\text{new}} = (\mathcal{V}^{\text{new}}, \mathbf{A}^{\text{new}}, \mathbf{X}^{\text{new}})$.

## 4.3 Training Objective

To effectively leverage both the topological structure learning and pseudo-anomaly classification capabilities, we design a joint training objective that combines regularization loss and classification loss. The overall loss function is formulated as:

$$\mathcal{L}_{\text{total}} = \alpha \cdot \mathcal{L}_{\text{reg}} + \beta \cdot \mathcal{L}_{\text{cls}}, \tag{9}$$

where $\alpha$ and $\beta$ are hyperparameters that balance the contributions of regularization and classification losses. The purpose of the regularization loss $\mathcal{L}_{\text{reg}} = \|Z\|_F^2$ is to prevent overfitting by penalizing the model for learning node embeddings $Z$ with excessively large values, where $Z$ represents the node embeddings learned by the encoder and $\| \cdot \|_F$ denotes the Frobenius norm. The classification loss is computed using a combination of the ground-truth labels and the refined pseudo-labels: $\mathcal{L}_{\text{cls}} = \text{BCE}(f(\mathbf{X}^{\text{new}}), \mathbf{Y}^{\text{new}})$, where $\text{BCE}(\cdot)$ is the binary cross-entropy loss function.

Table 1: Performance comparison of different GAD methods. We report GGAD and TAQ-GAD results under different label rates, with the highest scores bolded.

| Setting | Method | Dataset | | | | | | | | | |
|---|---|---|---|---|---|---|---|---|---|---|---|
| | | AUROC | | | | | AUPRC | | | | |
| | | Amazon | T-Finance | Reddit | Elliptic | Photo | Amazon | T-Finance | Reddit | Elliptic | Photo |
| Unsupervised | DOMINANT | 0.7025 | 0.6087 | 0.5105 | 0.2960 | 0.5136 | 0.1315 | 0.0536 | 0.0380 | 0.0454 | 0.1039 |
| | AnomalyDAE | 0.7783 | 0.5809 | 0.5091 | 0.4963 | 0.5069 | 0.1429 | 0.0491 | 0.0319 | 0.0872 | 0.0987 |
| | OCGNN | 0.7165 | 0.4732 | 0.5246 | 0.2581 | 0.5307 | 0.1352 | 0.0392 | 0.0375 | 0.0616 | 0.0965 |
| | AEGIS | 0.6059 | 0.6496 | 0.5349 | 0.4553 | 0.5516 | 0.1200 | 0.0622 | 0.0413 | 0.0827 | 0.0972 |
| | GAAN | 0.6513 | 0.3091 | 0.5216 | 0.2590 | 0.4296 | 0.0852 | 0.0283 | 0.0348 | 0.0436 | 0.0767 |
| | TAM | 0.8303 | 0.6175 | 0.6062 | 0.4039 | 0.5675 | 0.4024 | 0.0547 | 0.0437 | 0.0502 | 0.1013 |
| Semi-supervised | DOMINANT | 0.8867 | 0.6167 | 0.5194 | 0.3256 | 0.5314 | 0.7289 | 0.0542 | 0.0414 | 0.0652 | 0.1283 |
| | AnomalyDAE | 0.9171 | 0.6027 | 0.5280 | 0.5409 | 0.5272 | 0.7748 | 0.0538 | 0.0362 | 0.0949 | 0.1177 |
| | OCGNN | 0.8810 | 0.5742 | 0.5622 | 0.2881 | 0.6461 | 0.7538 | 0.0492 | 0.0400 | 0.0640 | 0.1501 |
| | AEGIS | 0.7593 | 0.6728 | 0.5605 | 0.5132 | 0.5936 | 0.2616 | 0.0685 | 0.0441 | 0.0912 | 0.1110 |
| | GAAN | 0.6531 | 0.3636 | 0.5349 | 0.2724 | 0.4355 | 0.0856 | 0.0324 | 0.0362 | 0.0611 | 0.0768 |
| | TAM | 0.8405 | 0.5923 | 0.5829 | 0.4150 | 0.6013 | 0.5183 | 0.0551 | 0.0446 | 0.0552 | 0.1087 |
| | CHRN | 0.9346 | 0.7581 | 0.5731 | 0.7315 | 0.6223 | 0.7865 | 0.0970 | 0.0500 | 0.2101 | 0.1420 |
| | SpaceGNN | 0.8616 | 0.8326 | 0.5010 | 0.6245 | 0.6314 | 0.5674 | 0.1942 | 0.0329 | 0.1257 | 0.1744 |
| | GGAD | 0.9443 | 0.8228 | 0.6354 | 0.7290 | 0.6476 | 0.7922 | 0.1825 | 0.0610 | 0.2425 | 0.1420 |
| | **TAQ-GAD** | **0.9474** | **0.8675** | **0.6682** | **0.7453** | **0.7107** | **0.7973** | **0.2255** | **0.0780** | **0.3573** | **0.2073** |

Table 2: Performance comparison on DGraph dataset. We report GGAD (G) and TAQ-GAD (T) results under different label rates, with the highest scores bolded.

| Metric | Baseline Methods | | | | | | GGAD (G) vs TAQ-GAD (T) | | | |
|---|---|---|---|---|---|---|---|---|---|---|
| | Unsupervised | | | Semi-supervised | | | 0.05% | 0.2% | 0.35% | 0.5% |
| | DOMINANT | AnomalyDAE | AEGIS | DOMINANT | AnomalyDAE | AEGIS | | | | |
| AUROC | 0.5738 | 0.5763 | 0.4509 | 0.5851 | 0.5866 | 0.4450 | G: 0.5892 | G: 0.5943 | G: 0.5902 | G: 0.5940 |
| | | | | | | | **T: 0.6327** | **T: 0.6569** | **T: 0.6581** | **T: 0.6623** |
| AUPRC | 0.0075 | 0.0070 | 0.0053 | 0.0076 | 0.0071 | 0.0058 | G: 0.0080 | G: 0.0082 | G: 0.0080 | G: 0.0083 |
| | | | | | | | **T: 0.0126** | **T: 0.0151** | **T: 0.0158** | **T: 0.0162** |

## 5 EXPERIMENTS

**Dataset and baseline methods.** We conduct experiments on six real-world graph datasets, including Amazon Dou et al. (2020), T-Finance Tang et al. (2022), Reddit Kumar et al. (2019), Elliptic Weber et al. (2019), Photo McAuley et al. (2015), and DGraph Huang et al. (2022b). More details about the datasets can be found in Appendix B. We adopt the semi-supervised learning scenario from GGAD Qiao et al. (2024), where ratios $\rho$ of the normal nodes in the training set are randomly sampled as labeled data, and the remaining ones are kept unlabeled. All anomaly nodes remain unlabeled during training. The data split and label rates are strictly consistent with those used in GGAD. To comprehensively evaluate the performance of TAQ-GAD, we select several representative GAD methods (DOMINANT Ding et al. (2019), AnomalyDAE Fan et al. (2020), TAM Qiao & Pang (2023), OCGNN Wang et al. (2021), AEGIS Ding et al. (2021), GAAN Chen et al. (2020), CHRN Gao et al. (2023), generative method GGAD Qiao et al. (2024)) as baselines. More details about the methods can be found in Appendix C.

**Evaluation metrics.** Our evaluation employs two standard metrics: AUROC and AUPRC Chai et al. (2022); Pang et al. (2021); Wang et al. (2023b). The AUROC score assesses the model's ability to distinguish between normal and anomalous instances overall. Given the inherent class imbalance in anomaly detection datasets, we also report AUPRC, which provides a more critical assessment of how well the model identifies true anomalies amidst the majority of normal nodes. For both metrics, the score ranges from 0 to 1, and a higher value signifies better detection performance.

**Implementation details.** We implement TAQ-GAD using PyTorch 1.10.0 and PyTorch Geometric within a Python 3.8 environment. The $\mathcal{N}(v_i)$ is defined as the $K$-hop neighborhood of node $v_i$. In TAQ-GAD, $K = 2$ for all datasets. The model employs a consistent embedding dimension of 300 across all datasets and is trained with the Adam optimizer. We rank all labeled nodes and select the top $\tau$-proportion of highest-scoring nodes as pseudo-anomaly nodes. A detailed analysis of key hyperparameters, including the pseudo-anomaly selection threshold $\tau$, the loss coefficients $\alpha$ and $\beta$,

Table 3: Ablation study results on different components of TAQ-GAD. We **bold** the highest scores.

| Component | Dataset | | | | | | | | | | | |
|---|---|---|---|---|---|---|---|---|---|---|---|---|
| | AUROC | | | | | | AUPRC | | | | | |
| | Amazon | T-Finance | Reddit | Elliptic | Photo | DGraph | Amazon | T-Finance | Reddit | Elliptic | Photo | DGraph |
| Baseline | 0.7262 | 0.6055 | 0.5733 | 0.3228 | 0.4198 | 0.5500 | 0.1595 | 0.0453 | 0.0448 | 0.0279 | 0.0889 | 0.0074 |
| +NBS | 0.8579 | 0.6931 | 0.6169 | 0.5022 | 0.7305 | 0.6047 | 0.6338 | 0.0540 | 0.0463 | 0.0368 | 0.2599 | 0.0157 |
| +PIS | 0.9033 | 0.7251 | 0.5754 | 0.6422 | 0.6114 | 0.5995 | 0.7277 | 0.1028 | 0.0422 | 0.0625 | 0.1386 | 0.0109 |
| +NBS + PIS | 0.9261 | 0.8261 | 0.6489 | 0.7159 | 0.7878 | 0.6577 | 0.7511 | 0.1298 | 0.0505 | 0.3979 | 0.4587 | 0.0166 |
| +NBS + PIS + TAE | **0.9571** | **0.8734** | **0.6873** | **0.7534** | **0.8632** | **0.6832** | **0.8230** | **0.2642** | **0.0631** | **0.4409** | **0.5942** | **0.0186** |
| $\mathcal{L}_{\text{cls}}$ | 0.9386 | 0.8360 | 0.6583 | 0.7315 | 0.8209 | 0.6627 | 0.7604 | 0.2235 | 0.0597 | 0.4076 | 0.5225 | 0.0157 |
| $\mathcal{L}_{\text{cls}} + \mathcal{L}_{\text{reg}}$ | **0.9571** | **0.8734** | **0.6873** | **0.7534** | **0.8632** | **0.6832** | **0.8230** | **0.2642** | **0.0631** | **0.4409** | **0.5942** | **0.0186** |

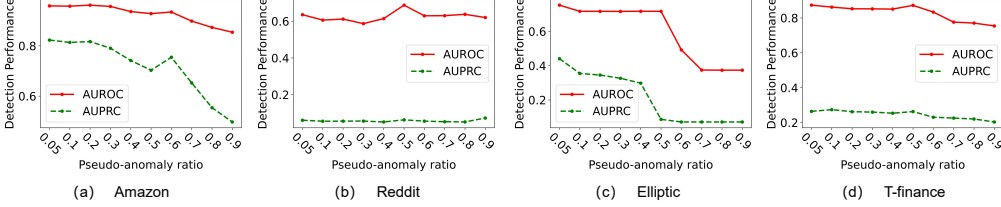

(a)  Amazon  (b)  Reddit  (c)  Elliptic  (d)  T-finance

Figure 3: Effect of $\tau$.

the normal node sampling ratio $\rho$, and the neighborhood hop count $K$, the $\lambda_1$ and $\lambda_2$ are provided in Section 5.3 and Appendix D. The source code of TAQ-GAD has been available on GitHub[1].

## 5.1 MAIN COMPARISON RESULTS

To evaluate the effectiveness of TAQ-GAD, we compare its performance against multiple baseline methods across various datasets. Following the experimental setup of GGAD, we randomly sample $\rho\%$ of the normal nodes as labeled normal data for training, where $\rho \in \{10, 15, 20, 25, 30\}$. The results of semi-supervised methods reported in the Table 1 are based on $\rho = 15$. Under this setting, our method outperforms all competing approaches.

To further validate the effectiveness of our approach, we compare TAQ-GAD with GGAD under different normal label rates across multiple datasets. At the $\rho = 30$, TAQ-GAD achieves significant performance improvements on several datasets. This performance advantage persists even as the label rate decreases. Under the $\rho = 20$, although GGAD demonstrates competitive performance on the T-Finance dataset (AUROC: 0.8560), TAQ-GAD still shows better or comparable results on other metrics and datasets. More importantly, under low-rate (15% and 10%), TAQ-GAD continues to stably outperform GGAD. At the $\rho = 10$, TAQ-GAD achieves higher AUROC scores on Amazon (0.9365 vs. 0.8796), T-Finance (0.8501 vs. 0.7252), Reddit (0.6367 vs. 0.6344), and Elliptic (0.7485 vs. 0.7301), while maintaining competitive AUPRC scores. Furthermore, on the challenging DGraph dataset with an extremely low anomaly rate (as shown in Table 2), TAQ-GAD consistently surpasses GGAD across all label rates (0.05% to 0.5%). At the 0.5% label rate, TAQ-GAD achieves an AUROC of 0.6623 and AUPRC of 0.0162, compared to GGAD's 0.5940 and 0.0083, demonstrating its superior capability in detecting rare anomalies under class imbalance. The consistent superiority of TAQ-GAD over GGAD across varying label rates and diverse dataset characteristics validates the effectiveness of our topological quantification approach.

## 5.2 ABLATION STUDY

To evaluate the contribution of each component in TAQ-GAD, we conduct an ablation study comparing the full model against several variants, with results summarized in Table 3. First, we examine the impact of the topological quantification modules. Using only a GCN as the baseline, we observe that incorporating either the NBS or PIS module individually leads to notable improvements in both AUROC and AUPRC across most datasets. When both modules are combined (+NBS+PIS), performance is consistently and significantly enhanced over the single-module configurations, confirming their complementary roles in capturing topological anomalies. Furthermore, the addition of the TAE module results in substantial gains across all datasets, achieving the highest scores in

---

[1]https://github.com/TingGuo301/TAQ-GAD

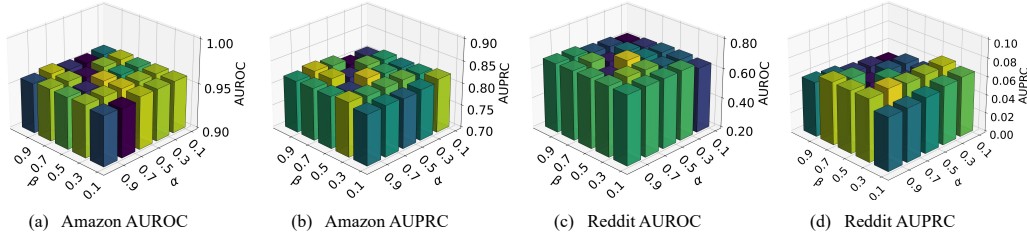

| (a) Amazon AUROC | (b) Amazon AUPRC | (c) Reddit AUROC | (d) Reddit AUPRC |

Figure 4: Effect of $\alpha$ and $\beta$.

every case. This demonstrates that TAE plays a critical role in strengthening anomaly connectivity through virtual anomaly centers, ultimately pushing the model to its optimal performance. More importantly, the classification loss $\mathcal{L}_{\text{cls}}$ serves as the primary objective, while the regularization loss $\mathcal{L}_{\text{reg}}$ constrains the node embeddings to prevent overfitting by penalizing excessively large values. Empirical results confirm that $\mathcal{L}_{\text{reg}}$ plays a critical role in the effectiveness of TAQ-GAD.

## 5.3 SENSITIVITY ANALYSIS

**The effect of $\tau$:** The sensitivity analysis of the pseudo-anomaly ratio $\tau$ (varied from 0.05 to 0.9) is shown in Figure 3. For Amazon, Elliptic, and T-Finance, performance peaks at $\tau = 0.05$ and declines with larger ratios, implying that excessive pseudo-anomalies introduce detrimental noise. This is particularly evident on the Elliptic dataset, where the AUROC exhibits a sharp decline from 0.7 to 0.4 as $\tau$ increases from 0.05 to 0.6, highlighting its pronounced susceptibility to noise interference. Conversely, Reddit's performance improves up to $\tau = 0.5$, indicating that its anomalous patterns require a broader pseudo-anomaly set for effective capture. Overall, a low $\tau$ of 0.05 is generally optimal, with performance degrading due to noise at higher values.

**The effect of $\alpha$ and $\beta$:** To evaluate the robustness of TAQ-GAD, we systematically vary the loss coefficients $(\alpha, \beta)$ from (0.1, 0.9) to (0.9, 0.1) across all datasets. The results, visualized in Figure 4, reveal a significant insensitivity to these hyperparameters. For instance, the Amazon dataset maintains robust performance under most weight combinations, indicating that our method does not require precise loss balancing to achieve optimal results. Similarly, Reddit exhibits minimal performance fluctuation.

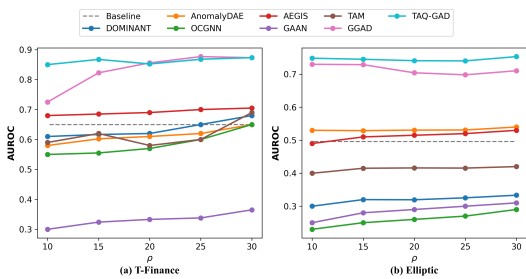

Figure 5: The effect of $\rho$.

**The effect of $\rho$:** As shown in Figure 5, we evaluated TAQ-GAD with different proportions of labeled normal nodes $\rho = \{10, 15, 20, 25, 30\}$. The results demonstrate that our model maintains excellent anomaly detection performance across all $\rho$ values. Notably, even with only $\rho = 10$, TAQ-GAD still outperforms all baseline methods. We believe this robust performance under limited supervision validates that our anomaly quantification module can reliably identify high-quality pseudo-anomalies, thereby enabling effective detection with minimal labeled data.

**The effect of $\lambda_1$ and $\lambda_2$:** In TAQ-GAD, $\lambda_1$ and $\lambda_2$ are hyperparameters for the NBS and PIS, respectively. NBS is assigned a higher weight as it captures behavioral patterns in local neighborhoods. PIS evaluates the structural isolation of nodes by measuring their connection strength. Setting an excessively high weight for PIS may introduce structural noise and interfere with the model's ability to detect genuine anomalies. To validate the robustness of parameter configuration, we systematically conducted sensitivity analysis of $\lambda_1$ and $\lambda_2$ across six datasets. As shown in Figure 6, experimental results demonstrate that our model maintains stable performance across four datasets, showing particular insensitivity to variations in $\lambda_2$ when $\lambda_1$ is fixed at 1.

## 5.4 COMPARISON WITH NAÏVE PSEUDO-LABELING STRATEGIES

To validate the effectiveness of the pseudo-label generation module in our proposed method, we present a detailed ablation study. TAQ-GAD is designed to select high-quality pseudo-anomalous

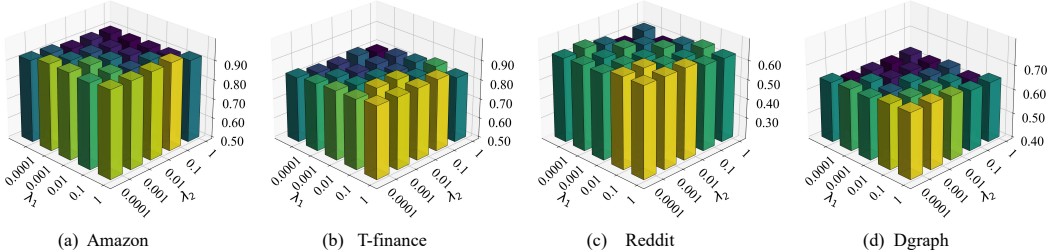

|         (a) Amazon | (b) T-finance | (c) Reddit | (d) Dgraph |

Figure 6: The effect of $\lambda_1$ and $\lambda_2$

nodes based on topological properties, which are subsequently used to train a model for obtaining sample-level pseudo-labels. We compare TAQ-GAD against two naive baseline strategies:

(1) **Random**: This baseline serves as a fundamental reference by randomly selecting a subset of nodes from the graph as pseudo-anomalous samples, without leveraging any structural or semantic information. These randomly selected nodes are then used to train the TAE module.

(2) **Low-degree**: This baseline selects nodes with relatively low degrees as pseudo-anomalous samples, based on the heuristic that anomalies often exhibit sparse connectivity or isolation from the majority of the graph. These low-degree nodes are subsequently used to train the TAE module.

Table 4: Performance comparison of different sampling strategies

|            | Reddit | | Photo | | Amazon | |
|------------|--------|--------|--------|--------|--------|--------|
|            | AUROC | AUPRC | AUROC | AUPRC | AUROC | AUPRC |
| Random     | 0.5443 | 0.0352 | 0.5953 | 0.1125 | 0.8980 | 0.7138 |
| Low-degree | 0.5579 | 0.0480 | 0.6359 | 0.1346 | 0.9084 | 0.6940 |
| TAQ-GAD    | **0.6682** | **0.0780** | **0.7107** | **0.2073** | **0.9474** | **0.7973** |
|            | T_finance | | Elliptic | | Dgraph | |
|            | AUROC | AUPRC | AUROC | AUPRC | AUROC | AUPRC |
| Random     | 0.7483 | 0.0907 | 0.6946 | 0.2228 | 0.4287 | 0.0101 |
| Low-degree | 0.7565 | 0.1119 | 0.7266 | 0.3423 | 0.3766 | 0.0092 |
| TAQ-GAD    | **0.8675** | **0.2255** | **0.7453** | **0.3573** | **0.6693** | **0.0178** |

Experiments are conducted across six standard benchmark datasets, with the results reported in Table 4. As shown in Table 4, the proposed topology-guided pseudo-label generation strategy (TAQ-GAD) significantly outperforms both baseline methods on the majority of datasets and evaluation metrics. For instance, on the Reddit dataset, TAQ-GAD improves AUROC by 22.8% and 19.8% over random selection and low-degree selection, respectively. These results demonstrate that TAQ-GAD's pseudo-label generation can more accurately identify potential anomalous instances. Besides, the low-degree selection strategy outperforms random selection on certain datasets (e.g., Elliptic), indicating that simple structural features do carry some discriminative information regarding anomalies, its performance remains substantially lower than that of our method. This validates that the comprehensive topological quantification employed in our approach exhibits stronger discriminative power compared to relying solely on node degree.

## 6 CONCLUSION

We propose TAQ-GAD, a generative graph anomaly detection framework designed to tackle the challenge of limited anomaly labels by leveraging topological structure analysis. The framework begins with a topological anomaly quantification module that employs two heuristic NBS and PIS metrics to identify high-quality pseudo-anomalies. Building upon this, the framework incorporates a TAE module. Importantly, the TAE module does not simply memorize these heuristic scores. Instead, starting from the pseudo-anomalous samples selected by NBS and PIS, it rectifies them through a risk-based pseudo-label generation mechanism and a label-flipping strategy. This process creates virtual anomaly centers and strengthens the topological connections among anomalies, enabling the module to learn more fine-grained node representations. Extensive experiments on six benchmark datasets demonstrate the superiority of TAQ-GAD over existing methods in terms of AUROC and AUPRC, highlighting its effective anomaly detection capabilities. Future research will develop priors for complex topological patterns, design scalable augmentation techniques, and extend the framework to dynamic graph anomaly detection with temporal modeling.

## ACKNOWLEDGE

This work was supported by the National Natural Science Foundation of China (62376141), the Natural Science Foundation of Shanxi Province, China (Grant No. 2024030212222153), the Shanxi Provincial Key Laboratory (CICIP2024002), the Shanxi Higher Education Institutions Science and Technology Innovation Project (2024L164).

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
