# A   PROOF OF THEOREM

**Notation** Let $G = (V, E)$ denote the graph, where nodes are partitioned into the normal set $\mathcal{V}_n$ and the anomalous set $\mathcal{V}_a$.

- $\pi_N = \frac{|\mathcal{V}_n|}{|\mathcal{V}|}$: proportion of normal nodes.

- $\pi_A = \frac{|\mathcal{V}_a|}{|\mathcal{V}|}$: proportion of anomalous nodes, with $\pi_N + \pi_A = 1$.

- $p_{v_x v_y}$: probability that a node of type $\mathcal{V}_x \in \{\mathcal{V}_n, \mathcal{V}_a\}$ connects to a node of type $\mathcal{V}_y \in \{\mathcal{V}_n, \mathcal{V}_a\}$ (Stochastic Block Model assumption).

- $\mathcal{N}(v_i)$: the 1-hop neighborhood of node $v_i$.

- $d_i$: degree of $v_i$.

- $E(\mathcal{N}(v_i))$: set of edges induced by $\mathcal{N}(v_i)$.

## A.1   THEOREM 1: EXPECTATION ANALYSIS OF NBS

**Theorem.** Under the stochastic block model (SBM) assumptions described above, the expected Node Boundary Score (NBS) of a normal node is strictly smaller than that of an anomalous node, i.e.,

$$\mathbb{E}[\text{NBS}(v_i) \mid v_i \in \mathcal{V}_n] < \mathbb{E}[\text{NBS}(v_i) \mid v_i \in \mathcal{V}_a].$$

**Proof.** For a node $v_i$ with neighborhood $\mathcal{N}(v_i)$, let

$$Z_{ij} = \mathbf{1}\{v_j \in \mathcal{N}(v_i) \cap \mathcal{V}_l\}$$

be the indicator variable that a neighbor $v_j$ is a labeled normal node. $\mathcal{V}_l \subset \mathcal{V}_n$ denotes the set of labeled normal nodes. Then the NBS score can be rewritten as

$$\text{NBS}(v_i) = 1 - \frac{1}{d_i} \sum_{v_j \in \mathcal{N}(v_i)} Z_{ij}, \qquad d_i = |\mathcal{N}(v_i)|.$$

Define $S_i = \sum_{v_j \in \mathcal{N}(v_i)} Z_{ij}$. Then

$$\mathbb{E}[\text{NBS}(v_i) \mid v_i \in \mathcal{V}_n] = 1 - \mathbb{E}\Big[\frac{S_i}{d_i} \,\Big|\, v_i \in \mathcal{V}_n\Big].$$

Applying the law of total expectation and conditioning on $d_i$, we obtain

$$\mathbb{E}\Big[\frac{S_i}{d_i} \,\Big|\, v_i \in \mathcal{V}_n\Big] = \mathbb{E}\Big[\mathbb{E}\Big[\frac{S_i}{d_i} \,\Big|\, d_i, v_i \in \mathcal{V}_n\Big] \,\Big|\, v_i \in \mathcal{V}_n\Big].$$

Given $d_i$ and $v_i$, the $d_i$ indicators $\{Z_{ij}\}$ are exchangeable. Hence, by linearity of expectation,

$$\mathbb{E}[S_i \mid d_i, v_i \in \mathcal{V}_n] = d_i \cdot \mathbb{P}(Z_{ij} = 1 \mid v_i \in \mathcal{V}_n),$$

which implies

$$\mathbb{E}\Big[\frac{S_i}{d_i} \,\Big|\, d_i, v_i \in \mathcal{V}_n\Big] = \mathbb{P}(Z_{ij} = 1 \mid v_i \in \mathcal{V}_n).$$

Therefore,

$$\mathbb{E}[\text{NBS}(v_i) \mid v_i \in \mathcal{V}_n] = 1 - \mathbb{P}(Z_{ij} = 1 \mid v_i \in \mathcal{V}_n).$$

It remains to compute this probability. Under the SBM, for a neighbor $v_j$ of $v_i \in \mathcal{V}_n$: - $v_j$ is normal with prior $\pi_N$, and then $(v_i, v_j)$ is an edge with probability $p_{v_n v_n}$; - $v_j$ is anomalous with prior $\pi_A$, and then $(v_i, v_j)$ is an edge with probability $p_{v_n v_a}$.

By Bayes' rule,

$$\mathbb{P}(Z_{ij} = 1 \mid v_i \in \mathcal{V}_n) = \frac{\pi_N p_{v_n v_n}}{\pi_N p_{v_n v_n} + \pi_A p_{v_n v_a}}.$$

Combining the above results, we obtain

$$\mathbb{E}[\text{NBS}(v_i) \mid v_i \in \mathcal{V}_n] = 1 - \frac{\pi_N p_{v_n v_n}}{\pi_N p_{v_n v_n} + \pi_A p_{v_n v_a}}.$$

A completely symmetric argument shows that

$$\mathbb{E}[\text{NBS}(v_i) \mid v_i \in \mathcal{V}_a] = 1 - \frac{\pi_N p_{v_a v_n}}{\pi_A p_{v_a v_a} + \pi_N p_{v_a v_n}}.$$

Finally, under the standard assumptions for anomaly detection that $p_{v_n v_n} > p_{v_a v_n}$ and $\pi_N > \pi_A$ (i.e., normal nodes form denser communities and anomalous nodes are rare), it follows that

$$\mathbb{E}[\text{NBS}(v_i) \mid v_i \in \mathcal{V}_n] < \mathbb{E}[\text{NBS}(v_i) \mid v_i \in \mathcal{V}_a].$$

This completes the proof.

### A.2 Theorem 2: Expectation Analysis of PIS

Recall that

$$\text{PIS}(v_i) = 1 - \frac{1}{2} \left( \frac{|E(\mathcal{N}(v_i))|}{\binom{d_i}{2} + \epsilon} + \frac{1}{D+1} \log(d_i + e) \right),$$

where $|E(\mathcal{N}(v_i))|$ denotes the number of edges within the induced subgraph of the neighbors of $v_i$. **Expected edge probability in the neighbor subgraph.** Let $v_i$ belong to class $\mathcal{V}_x \in \{\mathcal{V}_n, \mathcal{V}_a\}$. Then the distribution of neighbor types satisfies

$$\mathbb{P}(u_i \in \mathcal{V}_n \mid u_i \in \mathcal{N}(v_i), v_i \in \mathcal{V}_x) = \frac{\pi_N p_{v_x v_n}}{\pi_N p_{v_x v_n} + \pi_A p_{v_x v_a}},$$

$$\mathbb{P}(u_i \in \mathcal{V}_a \mid u_i \in \mathcal{N}(v_i), v_i \in \mathcal{V}_x) = \frac{\pi_A p_{v_x v_a}}{\pi_N p_{v_x v_n} + \pi_A p_{v_x v_a}}.$$

Hence, the expected probability that two neighbors $(u_i, w_i)$ are connected is

$$
\begin{aligned}
\mathbb{E}[\text{edge}(u_i, w_i) \mid v_i \in \mathcal{V}_x] = {} & \left( \frac{\pi_N p_{v_x v_n}}{\pi_N p_{v_x v_n} + \pi_A p_{v_x v_a}} \right)^2 p_{v_n v_n} \\
& + \left( \frac{\pi_A p_{v_x v_a}}{\pi_N p_{v_x v_n} + \pi_A p_{v_x v_a}} \right)^2 p_{v_a v_a} \\
& + 2 \cdot \frac{\pi_N p_{v_x v_n}}{\pi_N p_{v_x v_n} + \pi_A p_{v_x v_a}} \cdot \frac{\pi_A p_{v_x v_a}}{\pi_N p_{v_x v_n} + \pi_A p_{v_x v_a}} \cdot p_{v_n v_a}.
\end{aligned}
$$

**Expected number of edges in the neighbor subgraph.** Since the induced neighbor subgraph contains $\binom{d_i}{2}$ pairs of nodes, the expected number of edges is

$$\mathbb{E}[|E(\mathcal{N}(v_i))| \mid v_i \in \mathcal{V}_x] = \binom{d_i}{2} \cdot \mathbb{E}[\text{edge}(u_i, w_i) \mid v_i \in \mathcal{V}_x].$$

Thus,

$$\mathbb{E}\left[ \frac{|E(\mathcal{N}(v_i))|}{\binom{d_i}{2} + \epsilon} \,\middle|\, v_i \in \mathcal{V}_x \right] = \mathbb{E}[\text{edge}(u_i, w_i) \mid v_i \in \mathcal{V}_x] + O\left( \frac{1}{\binom{d_i}{2} + \epsilon} \right).$$

**Expectation of PIS.** Consequently, the expectation of PIS is

$$\mathbb{E}[\text{PIS}(v_i) \mid v_i \in \mathcal{V}_x] = 1 - \frac{1}{2} \left( \mathbb{E}[\text{edge}(u_i, w_i) \mid v_i \in \mathcal{V}_x] + \frac{1}{D+1} \log(d_i + e) \right) + o(1).$$

**Discriminability.** If $p_{v_n v_n} p_{v_a v_a} > p_{v_a v_n} p_{v_n v_a}$, then

$$\mathbb{E}[\text{edge}(u_i, w_i) \mid v_i \in \mathcal{V}_n] > \mathbb{E}[\text{edge}(u_i, w_i) \mid v_i \in \mathcal{V}_a].$$

Therefore,

$$\mathbb{E}[\text{PIS}(v_i) \mid v_i \in \mathcal{V}_n] < \mathbb{E}[\text{PIS}(v_i) \mid v_i \in \mathcal{V}_a],$$

which demonstrates that anomalous nodes exhibit strictly larger expected PIS values.

Table 5: Key statistics of the six datasets used in our experiments.

| Dataset | Type | # Nodes | # Edges | # Attributes | #Anomalies (Rate) |
|---------|------|---------|---------|--------------|-------------------|
| Amazon | Co-review | 11,944 | 4,398,392 | 25 | 821(6.9%) |
| T-Finance | Transaction | 39,357 | 21,222,543 | 10 | 1,803(4.6%) |
| Reddit | Social Media | 10,984 | 168,016 | 64 | 366(3.3%) |
| Elliptic | Bitcoin Transaction | 46,564 | 73,248 | 93 | 4,545(9.8%) |
| Photo | Co-purchase | 7,535 | 119,043 | 745 | 698(9.2%) |
| DGraph | Financial Networks | 3,700,550 | 73,105,508 | 17 | 15,509(1.3%) |

## B    DETAILED DATASET DESCRIPTION

Table 5 presents the key statistics of six widely adopted benchmark datasets for anomaly detection, covering network structures from diverse domains.

- **Amazon Dataset** Dou et al. (2020): A user-product review network based on the Musical Instrument category. The dataset labels users according to their helpful vote ratios, defining users with over 80% helpful votes as normal nodes and those with less than 20% helpful votes as anomalous nodes. The network encompasses various user relationships, with node features comprising 25 handcrafted attributes.

- **T-Finance Dataset** Tang et al. (2022): A financial transaction network where nodes represent anonymous accounts and edges denote transaction records between accounts. Node features include account attributes such as registration time, login behavior, and interaction frequency. Anomalous nodes primarily involve illegal activities like fraud money laundering and online gambling.

- **Reddit Dataset** Kumar et al. (2019): A user-subreddit interaction graph capturing user posting behaviors on Reddit over one month. Users banned by the platform are labeled as anomalous nodes. Node features are extracted from user post content through text mining techniques.

- **Elliptic Dataset** Weber et al. (2019): A Bitcoin transaction network where nodes represent Bitcoin transactions and edges indicate Bitcoin flow relationships. This dataset is specifically designed for identifying illicit transaction activities in blockchain networks.

- **Photo Dataset** McAuley et al. (2015): An Amazon co-purchase network where nodes represent products and edges indicate joint purchasing behaviors by users. Node features employ bag-of-words representations of user reviews.

- **DGraph Dataset** Huang et al. (2022b): A large-scale financial network containing millions of nodes and edges. Nodes represent user accounts in a financial company, while edges denote emergency contact relationships. Node features include demographic information such as age and gender, with users having overdue payment history labeled as anomalous.

These datasets exhibit diversity in scale, domain properties, and anomaly types, providing a comprehensive testing foundation for evaluating the performance of anomaly detection algorithms.

## C    MORE INFORMATION ABOUT THE COMPETING METHODS

### C.1    COMPETING METHODS

We select seven representative graph anomaly detection methods as comparative baselines. The technical characteristics of these methods are summarized as follows:

- **DOMINANT** Ding et al. (2019) employs an autoencoder architecture for graph anomaly detection. This method utilizes an encoder-decoder structure to simultaneously reconstruct both node features and graph structural information, combining feature reconstruction errors with structural reconstruction errors to form a comprehensive anomaly scoring mechanism.

- **AnomalyDAE** Fan et al. (2020) proposes a dual autoencoder framework that separately processes graph structural information and node attribute information, learning joint embedding representations of nodes and attributes in a latent space. The method incorporates an attention mechanism in the structural encoder to better capture normal structural patterns and improve anomaly detection performance.

- **OCGNN** Wang et al. (2021) integrates the concept of One-Class Support Vector Machine with graph neural networks, aiming to combine the recognition capability of one-class classifiers with the powerful representation learning ability of GNNs. It guides GNN training through a one-class hypersphere learning objective, identifying samples that fall outside the hypersphere as anomalous nodes.

- **AEGIS** Ding et al. (2021) designs specialized graph neural layers to learn anomaly-aware node representations and employs a generative adversarial network framework to detect anomalies in new data. The generator samples noise from a prior distribution to generate informative pseudo-anomalous samples, while the discriminator distinguishes between normal node representations and generated anomalous samples.

- **GAAN** Chen et al. (2020) constructs a generative adversarial network that generates fake graph nodes through a generator component. This method encodes nodes by computing sample covariance matrices for real and fake nodes, training a discriminator to identify whether two connected nodes originate from real or fake sources.

- **TAM** Qiao & Pang (2023) learns customized node representations by maximizing the local affinity between nodes and their neighbors, implementing an anomaly measure based on local affinity relationships. The method optimizes on truncated graphs by iteratively removing non-homophilic edges, ensuring that normal nodes exhibit significantly stronger local affinity than anomalous nodes in the learned representation space.

- **CHRN** Gao et al. (2023) explicitly bridges the heterophily in the spatial domain and the frequency in the spectral domain, and devise a label-aware edge indicator to calculate the post-aggregation similarity score based on which CHRN prunes possibly heterophily edges.

- **GGAD** Qiao et al. (2024) proposes a generative semi-supervised graph anomaly detection approach that generates pseudo anomaly nodes (outlier nodes) to provide effective negative samples for training a discriminative one-class classifier. The method leverages two important priors about anomaly nodes: asymmetric local affinity and egocentric closeness, generating outlier nodes that well assimilate real anomaly nodes in both graph structure and feature representation.

To effectively incorporate labeled normal samples into unsupervised frameworks, model-specific adaptations are implemented. For reconstruction-based approaches including DOMINANT and AnomalyDAE, the training process exclusively utilizes labeled normal nodes for data reconstruction. In OCGNN, the one-class center is optimized based solely on these normal instances. The TAM framework is trained by maximizing affinity specifically toward normal samples. Meanwhile, AEGIS and GAAN adopt a contrasting strategy by combining normal samples with synthetically generated anomalies to train an adversarial classifier. Remove the anomaly labels in CHRN and utilize the post-aggregation score to eliminate the interference of heterogeneous edge noise on the model.

## D ADDITIONAL EXPERIMENTAL RESULTS

### D.1 SENSITIVITY ANALYSIS

**The effect of $\alpha$ and $\beta$:** $\alpha$ and $\beta$ are hyperparameters that balance the contributions of regularization and classification losses, which operate at different scales and possess distinct characteristics. As systematically analyzed in Section 5.3 (Figure 4), the performance of TAQ-GAD exhibits notable insensitivity to the specific values of $\alpha$ and $\beta$. To reduce the overhead of hyperparameter tuning while maintaining performance, we further simplify the configuration by coupling the two parameters, fixing $\beta = 1 - \alpha$. As shown in Figure 7, experimental results confirm that even with this reduced parameterization, the model consistently achieves optimal performance across all datasets. This finding not only underscores the robustness of TAQ-GAD but also enhances its practical utility by easing the tuning process.

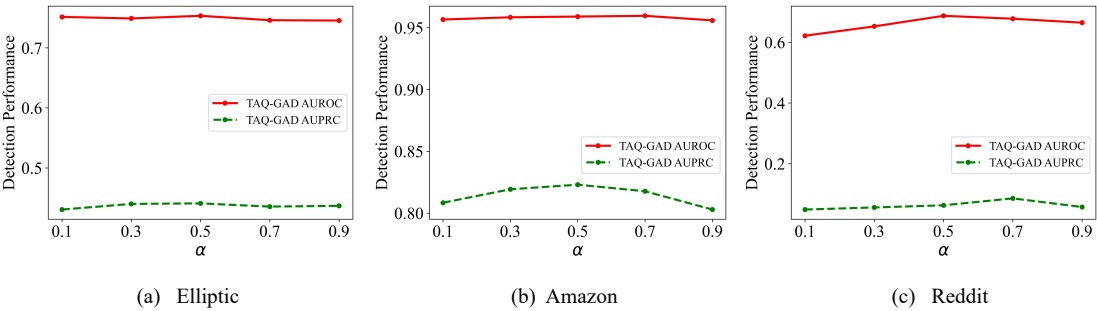

Figure 7: The effect of $\alpha$ and $\beta = 1 - \alpha$.

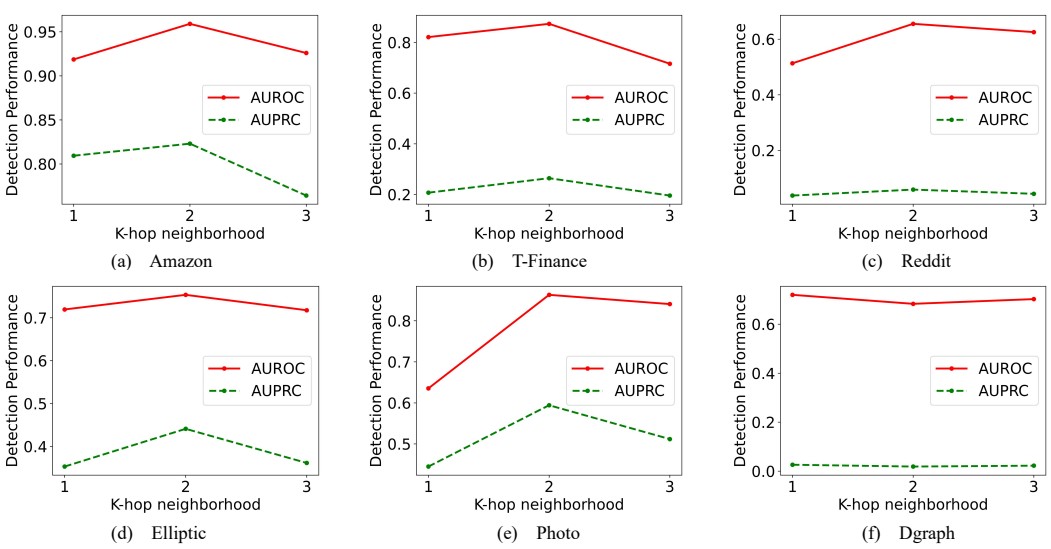

Figure 8: Performance under different $K$

**The effect of $K$:** As shown in Figure 8, the choice of $K$-hop neighborhood significantly impacts the performance. The experimental results demonstrate that most datasets achieve optimal performance at $K = 2$, a phenomenon with profound theoretical implications. When $K = 1$, the 1-hop neighborhood information is insufficient to capture complete anomaly patterns, as the topological characteristics of anomalous nodes often require a larger neighborhood range to be fully manifested. For instance, the AUROC of the Amazon dataset improves from 0.9186 at $K = 1$ to 0.9590 at $K = 2$, while T-finance shows an even more significant improvement from 0.8211 to 0.8734. However, when $K$ increases to 3, most datasets exhibit performance degradation, which occurs because the 3-hop neighborhood may introduce excessive noise that dilutes the discriminative power of local anomaly signals. An overly large neighborhood range incorporates too many normal nodes into the computation scope, causing the NBS and NIS indicators to lose sensitivity to genuine anomalous nodes.

**The effect of $\tau$:** Figure 9 presents the pseudo-anomaly ratio sensitivity analysis results on the remaining two datasets (Photo and Dgraph) to complement the findings reported in the main text.On the Photo dataset, TAQ-GAD achieves optimal performance at a 0.05 pseudo-anomaly ratio with AUROC of 0.86 and AUPRC of 0.59. Performance degrades significantly as the ratio increases. For the Dgraph dataset, the model similarly achieves best performance at a 0.05

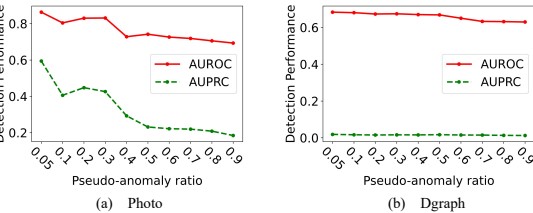

Figure 9: Effect of pseudo-anomaly ratio on detection performance.

Table 6: Comparison of Outlier Generation Methods

| Metric | Method | Dataset | | | | | |
|--------|--------|---------|-----------|--------|----------|--------|--------|
| | | Amazon | T-Finance | Reddit | Elliptic | Photo | DGraph |
| AUROC | Random | 0.7263 | 0.4613 | 0.5227 | 0.6856 | 0.5678 | 0.5712 |
| | NLO | 0.8613 | 0.6179 | 0.5638 | 0.6787 | 0.5307 | 0.5538 |
| | Noise | 0.8508 | 0.8204 | 0.5285 | 0.6786 | 0.5940 | 0.5779 |
| | GaussianP | 0.2279 | 0.6659 | 0.5235 | 0.6715 | 0.5925 | 0.5862 |
| | VAE | 0.8984 | 0.6674 | 0.6175 | 0.7055 | 0.6222 | 0.5801 |
| | GAN | 0.8288 | 0.5487 | 0.5378 | 0.6256 | 0.6032 | 0.5101 |
| | GGAD | 0.9324 | 0.8228 | 0.6354 | 0.7290 | 0.6476 | 0.5943 |
| | TAQ-GAD(ours) | **0.9571** | **0.8734** | **0.6873** | **0.7534** | **0.8632** | **0.6832** |
| AUPRC | Random | 0.1755 | 0.0402 | 0.0394 | 0.1981 | 0.1063 | 0.0061 |
| | NLO | 0.4696 | 0.1364 | 0.0495 | 0.1750 | 0.1092 | 0.0065 |
| | Noise | 0.5384 | 0.1762 | 0.0381 | 0.1924 | 0.1200 | 0.0076 |
| | GaussianP | 0.0397 | 0.0677 | 0.0376 | 0.1682 | 0.1194 | 0.0078 |
| | VAE | 0.6111 | 0.0652 | 0.0528 | 0.2344 | 0.1272 | 0.0063 |
| | GAN | 0.3715 | 0.0461 | 0.0433 | 0.1263 | 0.1143 | 0.0051 |
| | GGAD | 0.7843 | 0.1924 | 0.0610 | 0.2425 | 0.1442 | 0.0087 |
| | TAQ-GAD(ours) | **0.8230** | **0.2642** | **0.0631** | **0.4409** | **0.5942** | **0.0186** |

pseudo-anomaly ratio (AUROC: 0.68, AUPRC: 0.019). While the performance degradation is more gradual compared to Photo, both metrics show consistent decline as the pseudo-anomaly ratio increases.This substantial performance drop demonstrates that excessive pseudo-anomaly ratios introduce considerable noise that impairs detection accuracy.

## D.2 COMPARISON OF OUTLIER NODE GENERATION METHODS

To examine its effectiveness further, TAQ-GAD is also compared with other approaches that could be used as an alternative to generating the outlier nodes. These methods include: (i) **Random**, which randomly samples some normal nodes and treats them as outliers to train a class of discriminative classifiers; (ii) **Non-learnable Outliers (NLO)**, which generates outlier nodes by directly performing a simple aggregation (such as averaging) of the local neighbor representations of normal nodes, without performing learnable linear transformations and activations, and only relies on the original local structure statistics of normal samples to construct a parameter-free outlier initialization baseline; (iii) **Noise**, which directly generates the representation of outlier nodes from random noise; (iv) **Gaussian Perturbation (GaussianP)**, which directly adds Gaussian perturbations to the sampled normal node representations to generate outlier nodes. In addition to Noise and GaussianP, we further adopt two advanced generation methods: (v) **VAE**, which generates anomaly representations by reconstructing the original attributes of selected nodes, where our two anomaly prior-based constraints are applied to the generation process; (vi) **GAN**, which generates embedding representations from noise and adds an adversarial function to discriminate whether the generated nodes are fake or real, and (vii) **GGAD**, which optimizes the generation process by asymmetric local affinity and egocentric closeness, and finally uses a discriminative one-class classifier for anomaly detection. The results are shown in Table 6. **Random** method performs poorly because the randomly selected samples cannot be effectively distinguished from normal nodes, making it difficult to provide valuable anomaly references. **NLO** has performed well on some datasets such as Amazon and Elliptic, but there is still a significant gap compared to **TAQ-GAD**. Although the **Noise** and **GaussianP** methods can distinguish the generated anomaly point representations from normal nodes, due to the constraints of the graph structure, the generated results are seriously misaligned with the distribution of real anomaly nodes, and the final effect is limited. **VAE** and **GAN** have shown certain effects on some datasets, indicating that their prior mechanisms can assist in learning anomaly representations, but the overall performance of the two is still far inferior to **TAQ-GAD**. The abnormal nodes generated by **GGAD** perform better than some of the aforementioned methods by integrating the anomaly prior of graph structure and feature representation, but there are still shortcomings compared to **TAQ-GAD**, which fully demonstrates that **TAQ-GAD** can generate more accurate anomaly nodes by quantifying node abnormalities.

Table 7: Runtimes (in seconds) on the six datasets on CPU.

| Method | Amazon | T-Finance | Reddit | Elliptic | Photo | DGraph |
|---|---|---|---|---|---|---|
| DOMINANT | 1592 | 10721 | 125 | **1119** | 437 | **388** |
| AnomalyDAE | 1656 | 18569 | 161 | 8296 | 445 | 457 |
| OCGNN | 765 | 5717 | 162 | 3517 | 125 | / |
| AEGIS | 1121 | 15258 | 166 | 5638 | 417 | 1022 |
| GAAN | 1678 | 12120 | **94** | 1866 | 307 | / |
| TAM | 4516 | 17360 | 432 | 13200 | 165 | / |
| GGAD | 658 | 9345 | 368 | 5146 | **106** | 488 |
| TAQ-GAD (Ours) | **534** | **1832** | 134 | 2400 | 480 | 4214 |

# E  COMPUTATIONAL EFFICIENCY ANALYSIS

## E.1  TIME COMPLEXITY ANALYSIS

We analyze the time complexity of our method as follows:

**TAQ module.** The time complexity of the TAQ module is determined by the two scores: NBS requires $\mathcal{O}(|\mathcal{V}_l| * |\mathcal{N}(v_i)|)$ operations, while PIS requires $\mathcal{O}(|\mathcal{V}_l| * |E(\mathcal{N}(v_i))|)$ in the worst case. In practice, $|\mathcal{N}(v_i)|$ and $|E(\mathcal{N}(v_i))|$ are small and bounded for sparse graphs, such as social or citation networks, making our algorithm efficient and scalable.

**TAE Enhancement.** In the TAE module, the time complexities of node risk estimation and the label flipping strategy are $\mathcal{O}(|\mathcal{V}_l|)$ and $\mathcal{O}(|\mathcal{V}_l| * |\mathcal{N}(v_i)|)$, respectively. Since both scale linearly with the number of labeled nodes, the method is highly efficient and suitable for large-scale graphs.

## E.2  RUNTIME RESULTS

As shown in Table 7, our proposed TAQ-GAD demonstrates a competitive balance between computational efficiency and detection performance across most datasets. TAQ-GAD demonstrated excellent time efficiency on two larger datasets, Amazon and T-Finance. On the Amazon dataset, TAQ-GAD required only 534 seconds, saving at least 18% compared to other methods (compared to the next fastest, GGAD, which took 658 seconds). This advantage was even more pronounced on the T-Finance dataset, where TAQ-GAD achieved a time of 1832 seconds, significantly outperforming all other methods and saving approximately 68% of the computation time compared to the closest OCGNN (5717 seconds).

While TAQ-GAD's runtime was not the fastest on datasets like Reddit, Elliptic, Photo, and DGraph, the performance of the TAQ-GAD model demonstrates that this time cost is reasonable and worthwhile. For example, on the Reddit dataset, although GAAN achieved the fastest time (94 seconds), its AUROC was only 0.5216 (unsupervised) and 0.5349 (semi-supervised). TAQ-GAD, on the other hand, achieved an AUROC of 0.6873 (semi-supervised) in 134 seconds, representing a performance improvement of over 30%. Similarly, on the Photo dataset, the fastest GGAD achieved an AUROC of only 0.6476 in 106 seconds, while TAQ-GAD achieved an AUROC of 0.8632 in 480 seconds, a performance improvement of over 33%. This demonstrates that TAQ-GAD significantly improves anomaly detection accuracy while maintaining high computational efficiency, achieving a good balance between time efficiency and detection performance.

# F  USE OF LARGE LANGUAGE MODELS

Large language models were only used for minor language polishing. They were not involved in method design, theoretical analysis, or experiments, and therefore do not affect the originality of this work.