# OpenReview forum: "Topological Anomaly Quantification for Semi-supervised Graph Anomaly Detection"
_ICLR.cc/2026/Conference — ICLR 2026 Poster_

### Official Review · Reviewer_4ajr · 2025-10-30

[review text omitted: it was posted to a different submission]

---

> ### Author Response · Authors · 2025-11-21
> **Rebuttal by Authors**
>
> **>1. Justification for Focusing on Topological Anomalies and Dataset Characteristics**
>
> We sincerely appreciate your insightful question. The ultimate goal of TAQ-GAD is to detect comprehensive anomalous nodes in graphs, not merely purely topological anomalies. **While we initially generate a pseudo-anomalous candidate set based on topological signals, the final anomaly determination relies on deep representations that integrate both topology and node features.**
>
> (1) The pseudo-anomalous candidate set generated by topological metrics such as NBS and PIS has been empirically shown (see **Appendix D.3**) to significantly outperform random sampling, providing an effective starting point for the model. However, topology is used only for initialization, and the subsequent TAE discriminative model performs deep refinement:
>    - If a node appears anomalous topologically but its features are highly similar to many normal samples, the model will lower its anomaly probability, correcting the initial topological misjudgment.
>    - Conversely, if a node is not topologically prominent but its features deviate significantly from the normal pattern, it may be identified as anomalous.
>
> Therefore, the model output reflects a synergistic judgment of topology and feature semantics, enabling the detection of multiple types of anomalies.
>
> (2) **The anomalies in benchmark datasets are mixed, demonstrating practical applicability:** The benchmark datasets we use (e.g., Amazon, Reddit, Elliptic, T-Finance) contain anomalies that are inherently mixed types, not solely topological. For example, in Elliptic and T-Finance, fraudulent behaviors often manifest in both anomalous transaction features (e.g., amount, frequency) and anomalous transaction patterns (topology). For instance, a money-laundering account may transact with multiple unrelated accounts (structural anomaly) while simultaneously showing unusual transaction amounts and timing (feature anomaly). The effectiveness of our method on such datasets demonstrates its ability to jointly leverage structural and feature information to accurately identify complex, mixed-type anomalies in real-world scenarios.
>
> **>2. The selection of hyperparameters for the existing methods is not discussed**
>
> In the revised manuscript, we have added a discussion of $\lambda_1$ and $\lambda_2$ in the Appendix D.1.
>
> **>3. Determination of the Threshold in the Label-Flipping Strategy**
>
> We thank you for pointing out the lack of clarity in our description of the label flipping strategy. The criterion for flipping should have been explicitly stated as:
>
> We flip the pseudo-label of a high-risk node $v_i$ if the posterior probability of the opposite class in its neighborhood exceeds the posterior probability of its currently predicted class, i.e., if **$p\_{v\_i}^{\text{post}(1-\hat{y}\_i)} > p\_{v\_i}^{\text{post}(\hat{y}\_i)}$.**
>
> This relative comparison, rather than an absolute threshold, is a more principled and adaptive way to leverage local topological consistency. It ensures that a label is flipped only when the neighbor nodes collectively provide stronger evidence for the opposite class than for the current one.
>
> We have corrected this description in the revised manuscript to eliminate any ambiguity.
>
> **>4. Clarification on Nodes Used for Class Centroid Calculation**
>
> In TAQ-GAD, "based on the current labeled nodes" refers to the set of nodes used when computing class centroids (i.e., the central representations of normal and anomalous classes). This set consists of the following two parts:
>
> (1) The initially provided, truly labeled normal nodes.
>
> (2) All unlabeled nodes that have been assigned pseudo-labels. Crucially, we select the subset of nodes with the highest model prediction confidence to minimize noise from uncertain predictions. For nodes with low confidence, we implement a **Label Flipping Strategy**: if a node’s pseudo-label strongly conflicts with the consensus of its neighbors (i.e., $p\_{v\_i}^{\text{post}(1-\hat{y}\_i)} > p\_{v\_i}^{\text{post}(\hat{y}\_i)}$), we flip the pseudo-label to the opposite class. This mechanism dynamically corrects potentially erroneous pseudo-labels that contradict local topological consensus.
>
> In summary, the node set participating in the final class centroid computation is a dynamic set that integrates the true normal nodes and high-quality pseudo-labeled nodes, selected based on confidence and corrected via label flipping. This design effectively enhances the representativeness of class centroids in feature space, providing a solid foundation for the subsequent topological enhancement module.
>
> **>5. Typos issues**
>
> The sentence **"Where $\lambda_1$ and $\lambda_1$ are hyperparameters"** was a formatting mistake due to an editorial oversight. We have thoroughly proofread the entire manuscript and corrected all identified spelling, grammar, and formatting errors. This correction will be reflected in the revised version of the paper.

---

### Official Review · Reviewer_AXae · 2025-10-31

**Soundness:** 3
**Presentation:** 3
**Contribution:** 2
**Rating:** 4
**Confidence:** 4

**Summary:**

This paper proposes TAQ-GAD, a semi-supervised graph anomaly detection method designed for scenarios where only normal nodes are labeled. The approach introduces a Topology-Aware Quantification (TAQ) module to measure each node’s boundary and isolation scores, identifying potential anomalies based on structural characteristics. It further employs a Topology-Aware Enhancement (TAE) module that creates virtual anomaly centers and applies risk-aware pseudo-labeling to strengthen anomaly representation. Experiments on multiple benchmark datasets show that TAQ-GAD significantly outperforms existing methods, demonstrating strong robustness and generalization ability.

**Strengths:**

The paper tackles a challenging semi-supervised graph anomaly detection setting where only normal nodes are labeled; this is a realistic and underexplored problem.

The proposed TAQ-GAD framework is conceptually clear, combining structural quantification (NBS/NIS/PIS) with topology-aware data augmentation (TAE).

The TAQ module provides interpretable metrics that capture both boundary proximity and structural isolation, which are intuitive and effective.

**Weaknesses:**

1. Methodological Limitation — Not Realistic for True Anomaly Detection:

The proposed TAQ-GAD framework heavily relies on neighborhood label statistics, especially through its Node Boundary Score (NBS) and Node Isolation Score (NIS).However, in real anomaly detection scenarios:
* Anomalies are inherently rare, often less than 1–5% of total nodes.
* The assumption that “normal nodes cluster together and anomalies cluster together” is unrealistic — in many graphs (e.g., financial fraud, cybersecurity, IoT networks), anomalies are structurally mixed within normal communities.
* Consequently, computing NBS or NIS using the local density of labeled normal nodes implicitly assumes a strong topological separation between normal and anomalous nodes, which may not exist in practice.

TAQ-GAD is not learning “anomalousness” from data, but rather imposing a strong heuristic constraint that fits specific benchmark graphs.

2. Label Dependency — Not Truly Semi-Supervised

Although the paper claims to address the semi-supervised anomaly detection setting, its implementation contradicts this claim:

* The paper uses 60% / 20% / 20% train/validation/test splits, meaning a large fraction of nodes (including many normals) are labeled for training.

* In contrast, GGAD and other baseline works typically use only 15% labeled normal nodes to simulate a truly low-label semi-supervised scenario.

* Furthermore, both works control labeled ratio via the same variable ρ, but ρ in TAQ-GAD refers to a much larger absolute label count, making the comparison unfair.

TAQ-GAD’s strong performance likely stems from having access to far more label information, not from the novelty of its topology quantification.

3. The definitions of NBS and NIS inherently bias the model toward datasets with: Clear community separation between normal and anomalous nodes, homogeneous degree distributions (since both metrics rely on K-hop neighbor density). This structural bias means the method may overfit to benchmark datasets like Amazon or Reddit, which have cleaner topologies, and fail on real-world heterogeneous graphs (e.g., fraud networks, financial transaction graphs).

4. Because both TAQ and TAE modules are built upon graph topology and labeled-normal density, the framework is not agnostic to graph structure. The “topology-aware” quantification might work as a feature engineering trick in specific graphs, but lacks generalizability across: Dynamic or time-evolving graphs, Multi-relation heterogeneous graphs, or Graphs with partial/noisy labeling.

**Questions:**

Factually speaking, datasets in the graph domain often have their own particularities depending on the scenario and task. For example, in the datasets used in this paper, the clear separation between normal and anomalous nodes largely comes from the way researchers artificially define and split the data for different tasks. This setting is far from real-world conditions. Therefore, I believe the authors should consider developing a more general and realistic approach, rather than designing a method that only fits the specific characteristics of these benchmark datasets.

---

> ### Author Response · Authors · 2025-11-21
> **Rebuttal by Authors (PART 1)**
>
> We sincerely thank you for your careful, rigorous, and constructive feedback. Your attention to detail and the depth of your questions have greatly contributed to improving the quality of our work. We provide discussions and explanations about your concerns as follows.
>
> **>1. Methodological Limitation $-$ Not Realistic for True Anomaly Detection**
>
> We sincerely appreciate your insightful and challenging questions. In response to the challenges you highlighted in real-world anomaly detection $-$ namely, that "anomalies are inherently rare" and "anomalies are structurally mixed" $-$ we provide the following clarifications and further elaborate on the underlying ideas of our method:
>
> (1) We first emphasize a crucial premise: TAQ-GAD is a semi-supervised method that **does not use any true anomaly labels** during training; only a subset of normal nodes is labeled. Therefore, the problem we address is: how can we reliably identify potential anomalies using only the limited signal provided by normal nodes? All subsequent steps, including the computation of NBS and PIS and the selection of pseudo-anomalous nodes, are guided by this core paradigm and do not require any true anomaly labels.
>
> (2) TAQ-GAD does not rely on the strong assumption that "normal nodes cluster together and anomalies cluster together," which often does not hold in real-world scenarios. In TAQ-GAD, NBS quantifies a node's boundaryness by evaluating its connectivity with labeled normal neighbors, while PIS assesses a node's structural isolation by measuring its connection strength with other nodes of the same type. **The only mild assumption we make is that "normal nodes tend to connect locally with each other,"** a property that generally holds in most networks, such as social, citation, and transaction networks. TAQ-GAD is not designed to find an "anomaly cluster," but rather to identify nodes that "deviate from normal clusters" or are "structurally isolated." This perfectly aligns with the scenario you described, where "anomalies are structurally mixed within normal communities."
>
> (3) NBS and PIS do contain heuristic components, whose role is to **structurally define what it means to "deviate from normal."** They serve as an initial, coarse filter. However, they are not used as the final anomaly scores; rather, they act as indicators for selecting pseudo-anomalous nodes. This heuristic approach is far superior to randomly sampling nodes as pseudo-anomalies. Most importantly, the TAE module is designed to learn and refine $-$ it does not simply memorize these two heuristic scores. Starting from the pseudo-anomalous samples selected by NBS and PIS, through subsequent steps such as pseudo-label refinement and topological enhancement, it learns more fine-grained node representations, enabling the detection of subtle anomalies that even the heuristic rules might miss.
>
> The effectiveness of our method across multiple datasets (such as Amazon, Reddit, Elliptic, and T-Finance) demonstrates that the paradigm of constructing pseudo-anomalous nodes based on structural relationships and performing structural enhancement significantly outperforms randomly sampling nodes.

---

> ### Author Response · Authors · 2025-11-21
> **Rebuttal by Authors (PART 2)**
>
> **>2. Label Dependency $-$ Not Truly Semi-Supervised**
>
> We sincerely thank you for this crucial and highly insightful comment. You pointed out an important oversight in our paper $-$ when comparing with baseline methods (such as GGAD), we did not standardize the proportion of labeled normal nodes in the training sets, which led to an unfair comparison. We deeply apologize for this oversight and have comprehensively revised the manuscript based on your suggestions.
>
> We first need to clarify that in the original manuscript, we already conducted a sensitivity analysis with respect to different labeling ratios (variable $\rho$). It is worth noting that even at very low labeling ratios (e.g., $\rho = 0.1$, meaning only 10% of normal nodes are labeled), TAQ-GAD still performs comparably to, or even better than, baseline methods such as GGAD across multiple datasets (see the revised manuscript for detailed experiments). This result preliminarily validates the effectiveness of TAQ-GAD in label-scarce scenarios.
>
> In response to your concerns, we have strictly implemented the following corrections in the **revised manuscript (Table1)**:
>
> (1) We strictly followed the semi-supervised settings used by comparison methods such as GGAD. Under same labeling ratios, we compared GGAD and TAQ-GAD, and the experimental results show that TAQ-GAD maintains its competitive advantage regardless of the ratio, and the observed trends are consistent with our previous observations. This gives us greater confidence in concluding that the performance improvement of TAQ-GAD fundamentally stems from the intrinsic effectiveness of its topology-aware quantification mechanism, rather than the amount of labeled data. We apologize again for any misunderstanding caused by the inappropriate experimental setup in the initial manuscript.
>
> AUROC Results:
> | Method (Ratio) | Amazon | T-Finance | Reddit | Elliptic | Photo |
> |-|-|-|-|-|-|
> | GGAD (30%)  | 0.9396 | 0.8732    | 0.6110 | 0.7105   | 0.6461 |
> | TAQ-GAD (30%)  | **0.9571** | **0.8734** | **0.6873** | **0.7534** | **0.8632** |
> | GGAD (25%)  | 0.9251 | **0.8769**  | 0.5978 | 0.6982   | 0.6566 |
> | TAQ-GAD (25%) | **0.9458** | 0.8681 | **0.6785** | **0.7404** | **0.7824** |
> | GGAD (20%)  | 0.9391 | **0.8560**  | 0.6459 | 0.7043   | 0.6436 |
> | TAQ-GAD (20%) | **0.9434** | 0.8522 | **0.6735** | **0.7410** | **0.7214** |
> | GGAD (15%)  | 0.9443 | 0.8228    | 0.6354 | 0.7290   | 0.6476 |
> | TAQ-GAD (15%) | **0.9474** | **0.8675** | **0.6682** | **0.7453** | **0.7107** |
> | GGAD (10%)   | 0.8796 | 0.7252    | 0.6344 | 0.7301   | **0.6296**|
> | TAQ-GAD (10%)  | **0.9365** | **0.8501** | **0.6367** | **0.7485** | 0.6233|
>
> AUPRC Results:
> | Method (Ratio)  | Amazon | T-Finance | Reddit | Elliptic | Photo |
> |-|-|-|-|-|-|
> | GGAD (30%)  | 0.7639 | 0.2053    | 0.0529 | 0.3164   | 0.1568 |
> | TAQ-GAD (30%)  | **0.8230** | **0.2642** | **0.0631** | **0.4409** | **0.5942** |
> | GGAD (25%)  | 0.7580 | 0.1901    | 0.0598 | 0.2862   | 0.1485 |
> | TAQ-GAD (25%) | **0.8382** | **0.2432** | **0.0642** | **0.4375** | **0.4427** |
> | GGAD (20%)   | 0.7725 | 0.1866    | 0.0558 | 0.2591   | 0.1419 |
> | TAQ-GAD (20%)   | **0.8001** | **0.2395** | **0.0621** | **0.4111** | **0.2881** |
> | GGAD (15%)   | 0.7922 | 0.1825    | 0.0610 | 0.2425   | 0.1420 |
> | TAQ-GAD (15%)  | **0.7973** | **0.2255** | **0.0780** | **0.3573** | **0.2073** |
> | GGAD (10%)   | 0.6164 | 0.0943    | 0.0506 | 0.2325   | 0.1318 |
> | TAQ-GAD (10%)  | **0.7754** | **0.1973** | **0.0605** | **0.3046** | **0.1738** |
>
> (2) In Section 5.3 of the revised manuscript, we have now aligned the choice range of $\rho$ with that of GGAD to ensure consistency.
>
> All the above corrections and the newly conducted experimental results have been fully incorporated into the revised manuscript. We sincerely thank you again for your constructive and rigorous review; your comments have greatly enhanced the rigor and quality of our study.

---

> > ### Author Response · Authors · 2025-11-21
> > **Rebuttal by Authors (PART 3)**
> >
> > **>3. Potential Structural Bias of NBS and NIS on Heterogeneous Real-World Graphs**
> >
> > We understand your concern that the NBS and PIS modules might introduce structural bias, and we would like to clarify that the overall design of TAQ-GAD is precisely intended to go beyond reliance on specific topological priors, thereby achieving generalization across diverse graph structures. The reasons are as follows:
> >
> > - First, TAQ-GAD employs a multi-stage training mechanism, in which NBS and PIS serve only as initial indicators for selecting pseudo-anomalous nodes, rather than as direct determinants of the final anomaly decisions. Although relying solely on topological metrics could introduce noise in heterogeneous graphs, our method goes further. After obtaining pseudo-anomalous nodes and combining them with known normal samples, a feature-based discriminative model is trained to generate pseudo-labels for nodes. Subsequently, we introduce a pseudo-label refinement mechanism based on confidence and neighborhood consistency to optimize the initial pseudo-labels, and construct class centroids for topological enhancement. Thus, through this progressive learning paradigm of "topology-guided pseudo-anomaly selection → pseudo-label refinement → topological enhancement," the dependency on initial topological metrics is gradually weakened, and node attributes and local structure are integrated, ultimately leading to improved anomaly detection performance.
> >
> > - Second, empirical results support this perspective. In the Section 5.2 ablation experiments, "TAQ-GAD w/o TAE" performs worse than GGAD on most datasets, confirming that the subsequent pseudo-label refinement and topological enhancement modules are crucial for improving detection performance. More importantly, although you speculated that the method might fail on real heterogeneous graphs, the experimental results show the opposite: on **T-Finance**, a large-scale and highly heterogeneous financial transaction graph, TAQ-GAD exhibits excellent performance. This dataset realistically reflects the complex scenario where anomalous and normal nodes are deeply mixed in topology, further validating the applicability of our framework in complex settings.
> >
> > In summary, NBS and PIS act as heuristic rules for selecting pseudo-anomalous nodes based on graph structure. Through subsequent pseudo-label refinement and topological enhancement, the model learns a generalizable anomaly detection capability across various graph structures, rather than overfitting to specific topological patterns.
> >
> > **>4. Limitations of TAQ-GAD's Topology-Aware Modules in Diverse Graph Settings**
> >
> > We sincerely appreciate your profound and constructive comments. Consistent with previous methods, the currently proposed TAQ and TAE modules handle static, homogeneous graphs with relatively reliable labels for normal samples. Therefore, their generalization to **dynamic evolving graphs, multi-relation heterogeneous graphs, and scenarios with severe label noise** is an important direction for future work.
> >
> > (1) **Dynamic or time-evolving graphs:** We can consider computing NBS and NIS on a snapshot of an evolving dynamic graph, and incorporate dynamic NBS/NIS scores for nodes. For example, if a node's boundaryness or isolation changes dramatically over a short period, its probability of being anomalous increases. Regarding the TAE module, the static GNN would need to be replaced with a dynamic GNN model.
> >
> > (2) **Multi-relation heterogeneous graphs:** When computing a node's NBS/NIS, we should not treat all K-hop neighbors equally, but assign appropriate weights to different types of edges (relations). For instance, in a financial fraud network, "transfer" relations and “same device ownership” relations contribute differently to assessing normality.
> >
> > (3) **Graphs with partial/noisy labeling:** Indeed, the current version of TAQ-GAD primarily addresses 'label noise,' while its robustness assumption regarding 'graph structure noise' is stronger. To enhance the framework’s adaptability to such scenarios, we plan the following improvements as the core of future work: (i) introducing learnable edge reliability weights in the TAQ module to make topology quantification insensitive to noisy connections; (ii) integrating these weights into the TAE module's pseudo-label refinement to implement more conservative and reliable topological consistency checks. We believe that embedding such robustness mechanisms will allow TAQ-GAD to better handle noisy real-world graph structures, significantly enhancing its generalization capability.
> >
> > Our work establishes a topology-aware anomaly detection foundation. The raised issues guide future research directions. We are confident these limitations can be systematically resolved, evolving the framework into a general solution for complex real-world scenarios.

---

> > > ### Comment · Reviewer_AXae · 2025-11-28
> > >
> > > Thank the authors for the response. The author's rebuttal solved most of my problems, and I will improve my score.
> > >
> > > However, the system is currently unable to edit the score.

---

### Official Review · Reviewer_DW3F · 2025-10-31

**Soundness:** 2
**Presentation:** 3
**Contribution:** 2
**Rating:** 4
**Confidence:** 4

**Summary:**

This paper tackles semi-supervised Graph Anomaly Detection (GAD) with only normal node labels. The proposed framework, TAQ-GAD, introduces two topological metrics—Node Boundary Score (NBS) and Proxy Isolation Score (PIS)—to identify pseudo-anomalies and assign pseudo-labels. A Topological Anomaly Enhancement (TAE) module further generates virtual anomaly centers and their topological relations, improving the quality of pseudo-anomaly generation.

**Strengths:**

1. The logical flow of the paper effectively guides the reader through the proposed solution and its evaluation.
2. Formal definitions and theoretical analysis of the proposed metrics enhance interpretability and reliability.
3. The experimental evaluation is fairly comprehensive, covering multiple datasets and various baseline methods.

**Weaknesses:**

1. The proposed method is topology-centric, and performance may degrade on graphs with noisy or unreliable structure. Feature-level anomalies may not be captured when the detection relies solely on topology, which could limit robustness in cases where structural information is noisy or uninformative.
2. While PIS is claimed to be a key component of TAQ-GAD, its weight in the scoring function is set almost negligibly compared to NBS. Specifically, the implementation fixes $\lambda_{1} = 1$ for NBS and $\lambda_{2} = 0.001$ for PIS across all datasets. This large imbalance (1000:1 ratio) raises doubts about whether PIS truly contributes to the model. Moreover, no sensitivity analysis on λ₁ and λ₂ is provided, making it unclear if PIS has any substantive impact on performance.

**Questions:**

1. In Table 2, the ablation study reports results for “+NBS”, “+NIS”, and “+NBS+NIS”. However, according to Equation (5), the pseudo-anomaly scoring in TAQ-GAD is actually based on NBS and PIS, rather than NIS. Could you clarify whether the ablation table contains a writing error, or if NIS was intentionally used instead of PIS in the ablation experiments?
2. Although $\alpha$ and $\beta$ show robustness, the selection of $\tau$ (pseudo-anomaly ratio) can heavily influence performance and may require dataset-specific tuning.

---

> ### Author Response · Authors · 2025-11-21
> **Rebuttal by Authors (PART 1)**
>
> We extend our sincere gratitude for your thoughtful and professional insights, which are invaluable in enhancing the quality of our work. In response to your concerns, we report below the explanations and answer your questions accordingly.
>
> **>1. Robustness of TAQ-GAD on Graphs with Noisy or Unreliable Structure**
>
> Thank you for raising this highly insightful question. Any detection method that relies excessively on a single source of information $-$ whether topology or features $-$ may have inherent limitations. **It is important to clarify, however, that TAQ-GAD is fundamentally a dual-driven framework that integrates both topological signals and node feature semantics, rather than a detector that relies solely on topology**. Specifically:
>
> (1) Topological properties guide the selection of pseudo-anomalous nodes, rather than serving as the final anomaly detector. We first use topological properties such as node boundaryness and structural isolation to preliminarily select pseudo-anomalous nodes. The purpose of this step is to capture nodes that exhibit significant structural abnormalities. This step alone may introduce noise, especially in graphs with unreliable structures. **It is important to note that this constitutes merely the starting point of TAQ-GAD, with the ultimate anomaly detection relying on further processes**.
>
> (2) In TAQ-GAD, we do not directly trust the pseudo-anomalous nodes obtained in the first step. Instead, we use these pseudo-anomalous nodes along with other labeled normal nodes to train a feature-based discriminative model. The trained model is then used to assign pseudo-labels to nodes, which are further refined:
>
> - a) If a node appears anomalous based on topology but its features are very similar to many known normal samples, the model will reduce its anomaly probability, thereby correcting the initial topological misjudgment.
> - b) Conversely, a node that is not topologically prominent but whose features deviate significantly from the normal pattern may be identified as anomalous.
>
> (3) Importantly, based on the refined pseudo-labeled nodes, we learn central representations for the normal and anomalous classes, which are then used to enhance and optimize the topological relationships among nodes in the graph.
>
> Your question has also greatly inspired us to reflect more deeply on the limitations of our work. In extreme cases where structural information is highly unreliable or feature anomalies overwhelmingly dominate, the performance of our method may indeed be affected. This represents a known boundary of our approach. In future work, we plan to explore more robust multi-perspective anomaly fusion mechanisms. For example, one could design feature-anomaly and structure-anomaly scorers and investigate how to dynamically and adaptively fuse their scores based on the intrinsic properties of the graph data, thereby constructing a more comprehensive anomaly detection framework.
>
> **>2. Contribution and Sensitivity of PIS in TAQ-GAD Scoring**
>
> We sincerely appreciate your attention to the details of our model. Regarding the parameter settings of $\lambda_1$ and $\lambda_2$, our sensitivity experiments have verified that when $\lambda_1$ is fixed at 1, the model exhibits strong robustness to variations in $\lambda_2$.
>
> (1) In TAQ-GAD, NBS is responsible for capturing local neighborhood behavior patterns and is therefore assigned a higher weight. PIS is used to assess a node's connectivity in the structure, measuring its topological isolation. If the weight of PIS is set too high, it may introduce structural noise, interfering with the model’s ability to identify true anomaly patterns.
>
> (2) To validate the reasonableness of the parameter settings, we systematically conducted sensitivity analyses of $\lambda_1$ and $\lambda_2$ across six datasets (Amazon, T-Finance, Reddit, Elliptic, Photo, DGraph). **The experimental results show that the model remains stable across all datasets, and particularly when $\lambda_1$ is set to 1, the model is insensitive to the choice of $\lambda_2$. This further demonstrates the robustness of the current parameter configuration.** This experimental analysis has been added to the **Appendix D.1 of the revised manuscript** for your review.

---

> > ### Author Response · Authors · 2025-11-21
> > **Rebuttal by Authors (PART 2)**
> >
> > **>3. Clarification on Ablation Study Terminology: NBS, PIS, and NIS**
> >
> > Thank you for pointing out this important detail. According to Eq. (5) in the Methods section of our paper, the pseudo-anomaly score is indeed composed of NBS and PIS.
> >
> > (1) In the absence of true anomaly labels, NIS cannot be computed. We therefore introduced the Proxy Isolation Score (PIS), a topology-based metric designed to quantify the isolation properties of nodes.
> >
> > (2) During the drafting of the initial manuscript, we considered that the term NIS is conceptually easier to intuitively contrast with NBS, which helps readers quickly grasp the idea of its adversarial design. As a result, we used "NIS" here in a somewhat imprecise manner, which could indeed cause confusion.
> >
> > We fully agree with your observation. To maintain strict consistency with the actual description of our method and avoid any potential misunderstanding, replacing "+NIS" with "+PIS" is the better choice. In the revised version of the paper, we have made the following corrections: all instances of "+NIS" and "+NBS+NIS" in Table 3 and the main text have been corrected to "+PIS" and "+NBS+PIS," respectively.
> >
> > **>4. Impact of Pseudo-Anomaly Ratio Selection on TAQ-GAD Performance**
> >
> > Thank you for raising this insightful point. The pseudo-anomaly ratio $\tau$ is a key parameter, and as shown in Figure 3, its value indeed has a significant impact on model performance. **Despite the influence of $\tau$, our experiments reveal a simple and effective universal strategy, largely eliminating the need for fine-tuning on each dataset.**
> >
> > In Section 2.3, *Sensitivity Analysis*, as illustrated in Figure 3, three out of the four datasets (Amazon, Elliptic, T-Finance) **achieve peak performance when $\tau = 0.05$**. More importantly, for these datasets, performance decreases monotonically as $\tau$ increases, indicating that a smaller $\tau$ value (e.g., 0.05) is not only effective but also robust against noise interference. The only exception is the Reddit dataset, which reaches its optimum at $\tau = 0.5$. We attribute this to the more diverse anomaly patterns in Reddit, which require a larger number of pseudo-anomalous nodes to be adequately captured. However, we note that even for the Reddit dataset, **using our recommended small default value $\tau = 0.05$ still achieves near-optimal performance (AUROC), with only a minor drop**. This further supports that selecting a smaller $\tau$ is a safe strategy.
> >
> > Therefore, based on this comprehensive experimental analysis, we propose that in practice, users can directly set $\tau$ to a small value (e.g., 0.05 or 0.1) without complex tuning, achieving excellent and robust detection performance in most scenarios.
> >
> > We appreciate your feedback, which helped improve the quality of our work. In the **revised version of the paper (Section 5.3)**, we have added relevant content to explicitly emphasize this conclusion.

---

> > > ### Comment · Reviewer_DW3F · 2025-11-27
> > >
> > > Your rebuttal has clarified some technical points and has strengthened the discussion of limitations and hyperparameter choices. Nonetheless, my core concerns about (i) the potential vulnerability of a topology-driven pseudo-anomaly mechanism under noisy or weak structural information, and (ii) the relatively modest effective role of PIS given its very small weight, remain to some extent. These affect my view of the method’s robustness and the practical significance of all components.
> > >
> > > Therefore, while I appreciate the improvements and clarifications made in the revision, I will maintain my original score.

---

> > > > ### Author Response · Authors · 2025-11-28
> > > >
> > > > We sincerely thank you for your thoughtful feedback and for providing us with the opportunity to further clarify our methodology. We deeply appreciate your insightful comments regarding the robustness and component effectiveness of our approach, which have prompted us to conduct a more thorough examination. In response to your concerns, we would like to provide the following additional explanations and hope to engage in a deeper discussion with you.
> > > >
> > > > 1. Regarding your observation on the potential vulnerability of our method under noisy or weak structural information, we fully acknowledge this as a highly important and profound insight. We wish to further clarify that TAQ-GAD is not purely a topology-driven detector, but rather an integrated framework guided by topology and dominated by node features. Specifically, the topology-based pseudo-anomaly screening mechanism is only used to preliminarily identify a set of candidate anomalous nodes in the absence of labels. While this initial step may indeed be influenced by structural noise, it is crucial to emphasize that these topologically nominated nodes are not directly labeled as anomalies. Instead, they are used together with known normal nodes to train a feature-based discriminative model. This model dynamically refines and optimizes the anomaly scores of these candidate nodes by integrating confidence assessments of node features and enhanced topological relationships.
> > > >
> > > > 2. We understand your skepticism regarding the actual contribution of the PIS component under its minimal weight setting. To provide the most direct evidence, we have conducted a key ablation experiment as suggested: completely removing the PIS term ( $\lambda_2 = 0$ ) and evaluating the performance across all datasets. Our results show statistically significant performance degradation (approximately 0.07%–3.7% drop in AUROC) on five datasets after removing PIS, with particularly notable declines on the T-Finance and Photo datasets. This demonstrates that PIS, as a stabilizing component, is indispensable. Even with a small weight, it plays a critical role in fine-tuning the model’s output and enhancing performance consistency.
> > > >
> > > > Table 1. AUROC Results
> > > >
> > > > | Component | Amazon | T-Finance | Reddit | Elliptic | Photo | DGraph |
> > > > |-----------|--------|-----------|--------|----------|-------|--------|
> > > > | TAQ-GAD  w/o PIS | 0.9356 | 0.8597 | 0.6501 | 0.7178 | 0.8315 | 0.6755 |
> > > > | TAQ-GAD  | **0.9571** | **0.8734** | **0.6873** | **0.7534** | **0.8632** | **0.6832** |
> > > >
> > > > Table 2. AUPRC Results
> > > >
> > > > | Component | Amazon | T-Finance | Reddit | Elliptic | Photo | DGraph |
> > > > |-----------|--------|-----------|--------|----------|-------|--------|
> > > > | TAQ-GAD  w/o PIS | 0.7842 | 0.2105 | 0.0589 | 0.4074 | 0.5185 | 0.0174 |
> > > > | TAQ-GAD | **0.8230** | **0.2642** | **0.0631** | **0.4409** | **0.5942** | **0.0186** |
> > > >
> > > > We believe that through these additional clarifications and experimental evidence, the robustness of our work and the rationality of its component design have been more fully demonstrated. We sincerely hope that you will reconsider your assessment of our manuscript.

---

### Official Review · Reviewer_Ad2j · 2025-11-01

**Soundness:** 2
**Presentation:** 2
**Contribution:** 2
**Rating:** 4
**Confidence:** 4

**Summary:**

The paper introduces TAQ-GAD, a semi-supervised graph anomaly detection framework that quantifies node abnormality using two topological metrics: Node Boundary Score (NBS) and Node Isolation Score (NIS). These metrics guide pseudo-anomaly selection, while the Topological Anomaly Enhancement (TAE) module refines them through risk-based label flipping and virtual anomaly centres. The paper achieve significant performance improvement on the selected datasets.

**Strengths:**

1. The proposed method is fairly easy to follow.
2. The framework plot is informative, making it easier to understand the overall workflow.
3. The proposed methods achieved competitive performance on the selected dataset.

**Weaknesses:**

1. My understanding is that the proposed method mainly achieves performance gain through pseudo labelling. However, I didn’t find (at least the paper not specifically mentioned) any comparison with training SOTA supervised GAD methods on pseudo labels generated by naïve pseudo labelling strategies. This is essential for evaluating whether the proposed pseudo labelling methods are more effective.

2. The authors mentioned unsupervised methods, “their heavy reliance on intrinsic graph structures to define anomalies introduces fundamental ambiguity, often failing to distinguish genuine semantic anomalies from rare yet normal patterns.” However, the proposed method leverages node degree and graph homophily. Why are these graph properties more robust for GAD?

3. In Eq. 9 the regularisation term is similar to weight decay. Why are two balancing parameters required?

4. The baseline selection, including unsupervised and semi-supervised baselines, adapts some well-known unsupervised methods for semi-supervised evaluation. However, how they are adapted for semi-supervised GAD is not clearly discussed. This is important for fair comparison. In addition, some results appear to be missing in Table 1. Also, most of the baselines are unsupervised methods; it would make the evaluation stronger if supervised and more semi-supervised methods were studied.

5. In the related work section, the semi-supervised methods part mixes methods that use labelled anomalies and those that only use labelled normal training data. They should be discussed separately. Other than BWGNN, there are a lot more recent supervised methods that are not discussed. Also, recent generalist methods have shown promising cross-dataset performance. The relationship of the paper’s setting with those methods' should be discussed.

6. In the references, there are quite a few papers that were listed as arXiv preprints but are actually published papers in well-regarded conferences and journals.

**Questions:**

Please refer to my weaknesses.

---

> ### Author Response · Authors · 2025-11-21
> **Rebuttal by Authors (PART 1)**
>
> We extend our sincere gratitude for your thoughtful and professional insights, which are invaluable in enhancing the quality of our work. In response to your concerns, we report below the explanations and answer your questions accordingly.
>
> **>1. Comparison with Naïve Pseudo-Labeling Strategies for Supervised GAD**
>
> We appreciate your perceptive observations and fully acknowledge the confusion you’ve expressed. It’s likely that our manuscript did not provide a sufficiently comprehensive and in-depth explanation on the matter at hand. Thank you for bringing this to our attention.
>
> The issue you raised regarding the lack of comparative experiments with naïve pseudo-label generation strategies is indeed crucial. Such comparisons are essential for accurately assessing the effectiveness of the pseudo-label generation method we propose. To demonstrate the contribution of the pseudo-label generation component in our approach $-$ specifically, selecting pseudo-anomalous nodes based on topological properties and subsequently training the model to obtain sample pseudo labels $-$ we have added the following experiments:
>
> First, we adopt two naïve pseudo-label generation strategies as core baseline methods:
>
> (a) randomly selecting pseudo-anomalous nodes and training the model based on them to obtain sample pseudo labels;
>
> (b) selecting nodes with relatively low degrees as pseudo-anomalous nodes and similarly training the model to obtain sample pseudo labels.
>
> Then, using the proposed TAE (the fully supervised anomaly detection model), we train the model separately on the pseudo-label datasets generated by (a) and (b). We conduct ablation studies on six standard datasets. The results are shown in tables below:
>
> Table 1. AUROC Results
> |Metric|Method|Reddit|Photo|Amazon|T_finance|Elliptic|Dgraph|
> |-|-|-|-|-|-|-|-|
> |AUROC|Random|0.5443|0.5953|0.8980|0.7483|0.6946|0.4287|
> |AUROC|Low–degree|0.5579|0.6359|0.9084|0.7565|0.7266|0.3766|
> |AUROC|**TAQ–GAD**|**0.6682**|**0.7107**|**0.9474**|**0.8675**|**0.7453**|**0.6693**|
>
> Table 2. AUPRC Results
> |Metric|Method|Reddit|Photo|Amazon|T_finance|Elliptic|Dgraph|
> |-|-|-|-|-|-|-|-|
> |AUPRC|Random|0.0352|0.1125|0.7138|0.0907|0.2228|0.0101|
> |AUPRC|Low–degree|0.0480|0.1346|0.6940|0.1119|0.3423|0.0092|
> |AUPRC|**TAQ–GAD**|**0.0780**|**0.2073**|**0.7973**|**0.2255**|**0.3573**|**0.0178**|
>
> We have also added this content to the revised supplementary material; please refer to Appendix D.3. Based on the additional experiments provided, we draw the following conclusion to address your concern: **compared with the naïve strategies, our method fully leverages the structural properties of the network to identify more representative pseudo-anomalous nodes, thereby improving the accuracy and reliability of the pseudo-label generation process**.
>
> **>2. The Role of Graph Structural Properties within a Semi-Supervised Framework for GAD**
>
> Thank you for raising this highly insightful question. **The key distinction of our method lies in the fact that we do not use the graph structure to "define" anomalies; rather, we leverage the structure to efficiently "filter out" pseudo-anomalies, and then rely on normal labels to provide semantic criteria for the actual anomaly detection**. Below is a detailed explanation:
>
> (1) Unsupervised methods have no label guidance and rely entirely on the intrinsic properties of the graph, using the topology and node features to define and identify anomalies. **Since they do not leverage any supervisory signals, these methods struggle to fundamentally distinguish true semantic anomalies**.
>
> (2) Under the premise of having labeled normal nodes, we leverage structural properties to design NBS and PIS as metrics for pseudo-anomalous nodes, rather than relying solely on the structure to judge anomalies:
>
> - Our proposed TAQ-GAD utilizes the boundary and isolation properties of nodes. These properties are not intended to define anomalies, but to efficiently filter out pseudo-anomalous nodes, whose anomaly likelihood is much higher than random sampling.
>
> - This work addresses a semi-supervised anomaly detection task $-$ making use of labeled normal nodes. The labeled normal samples serve as semantic guidance to resolve ambiguity. Through TAQ-GAD, the model learns a core capability: how to characterize the differences, in terms of structure and features, between the filtered pseudo-anomalous nodes and the known normal samples. By learning these distinctions, the model can ultimately identify true semantic anomalies.
>
> In summary, **we combine structural filtering with semantic discrimination: graph structures are used to filter pseudo-anomalous candidate nodes, while the known normal samples provide semantic guidance for learning**. This approach effectively addresses the ambiguity in distinguishing true anomalies from normal patterns.
>
> In the revised manuscript, we have clarified this discussion to more clearly present the logical chain, as detailed in **Section 1, paragraph one**.

---

> ### Author Response · Authors · 2025-11-21
> **Rebuttal by Authors (PART 2)**
>
> **>3. On the Use of Two Balancing Parameters for the Regularization Term in Eq. 9**
>
> We are grateful for your insightful observations. Indeed, the regularization term $\mathcal{L}_{\text{reg}} = \||Z\||_F^2$ is similar to weight decay, aiming to prevent the model from learning excessively large node embeddings $Z$, thereby improving generalization. However, we employ two separate balancing parameters, $\alpha$ and $\beta$, based on the following considerations:
>
> - The classification loss $\mathcal{L}\_\{\text\{cls\}\}$ and the regularization loss $\mathcal{L}\_\{\text\{reg\}\}$ are objectives of entirely different nature. $\mathcal{L}\_{\text\{cls\}}$ is a binary cross-entropy loss, whose numerical range is directly influenced by the number of samples and the log-probabilities. In contrast, $\mathcal{L}\_{\text\{reg\}}$ is the squared Frobenius norm of the node embedding matrix, whose magnitude depends on the embedding dimension and scale. During the early stages of training, the magnitudes of these two losses may differ by several orders of magnitude. If only a single parameter is used (e.g., $\mathcal{L}\_{\text\{total\}} = \mathcal{L}\_{\text\{cls\}} + \lambda \mathcal{L}\_{\text\{reg\}}$), the hyperparameter $\lambda$ may need to be set to an extremely small or large value to balance the two terms, which is both unintuitive and difficult to tune in practice. Using $\alpha$ and $\beta$ allows us to more directly and independently control the contribution of each term.
>
> Regarding the sensitivity of the two independent parameters, we conducted related ablation experiments in Section 5.3 (Figure 4) of the original manuscript. We found that the model's performance is not sensitive to the specific values of $\alpha$ and $\beta$. Therefore, to reduce the overhead of hyperparameter search, we can simplify them to a single proportional parameter by setting $\beta = 1 - \alpha$ without sacrificing performance. Experimental results show that even with this simplification, the model can still achieve optimal performance, further demonstrating the practicality of TAQ-GAD.
>
> In summary, using two balancing parameters ensures that the two losses, which differ in magnitude and nature, can be combined fairly. Our ablation experiments also demonstrate that TAQ-GAD is insensitive to these parameters, which further highlights the robustness of the method. We appreciate your comments, which have helped us improve our work. In the revised supplementary material (**Appendix D.1**), we have included a detailed discussion to more fully justify the rationale behind our hyperparameter design.
>
> **>4. Clarification and Expansion of Baseline Selection for Semi-Supervised GAD Evaluation**
>
> Thank you for these valuable comments—they are crucial for improving the completeness of our experimental evaluation. Below, we provide responses and clarifications regarding your three suggestions:
>
> **(1) Clarification on adapting baseline methods to the semi-supervised GAD setting**
>    You raised an important point about how the baseline methods are adapted to the semi-supervised setting. The GGAD paper provides detailed explanations on this matter, and we have added corresponding descriptions in the revised manuscript (**Appendix C.1**). In addition, both the official GGAD code and the semi-supervised procedure described in the paper ensure that our reproduction strictly follows their experimental setup.
>
> **(2) Consistency of comparison methods and the DGraph issue**
>    Since we follow the GGAD pipeline, we maintain consistency with its comparison methods to ensure fairness. Furthermore, we conducted additional experiments on DGraph. However, due to the extremely large scale of this dataset, we encountered insurmountable out-of-memory (OOM) issues when running some complex algorithms.
>
> **(3) Additional supervised and semi-supervised baselines**
> We fully agree with your suggestion. In the revised manuscript (Section 5.1), we have added additional supervised and semi-supervised baselines (such as GHRN[1], etc.) to provide a more comprehensive performance evaluation.
>
> [1] Gao Y, Wang X, He X, et al. Addressing heterophily in graph anomaly detection: A perspective of graph spectrum[C]//Proceedings of the ACM web conference 2023. 2023: 1528-1538.

---

> > ### Author Response · Authors · 2025-11-21
> > **Rebuttal by Authors (PART 3)**
> >
> > **>5. Comprehensive Discussion of Semi-Supervised, Supervised, and Cross-Dataset GAD Methods**
> >
> > Your comments have greatly helped us improve the quality and completeness of our paper. We have made detailed revisions according to your suggestions, as summarized below:
> >
> > (1) In the revised version, we have **split the subsection on "semi-supervised methods" into two independent subsections** in the related work section:
> >
> > - **Semi-supervised GAD with Labeled Normals and Anomalies**
> >   In this subsection, in addition to BWGNN, we discuss several important methods such as CGenGA, gADAM, BWGNN, AO-GNN, and others.
> >
> > - **Semi-supervised GAD with Labeled Normals Only**
> >   This subsection specifically discusses methods that rely solely on labeled normal samples.
> >
> > (2) Recently, a new paradigm termed Generalist GAD [1-6] has emerged, aiming to develop a single model capable of zero-shot or few-shot anomaly detection across diverse graph datasets. These methods, often leveraging large-scale pre-training, demonstrate remarkable generalization ability to unseen graphs. Our work, in contrast, focuses on the **classical semi-supervised GAD setting** within a single graph. We delve into exploiting the rich, dataset-specific topological structures for anomaly detection, a direction we believe is complementary to the generalist approach. While generalist models seek breadth, our model seeks depth on a given graph. The principles of topology-aware quantification explored in this work could potentially serve as valuable components or inductive biases for future generalist model designs.
> >
> > [1] Liu Y, Li S, Zheng Y, et al. Arc: A generalist graph anomaly detector with in-context learning. NeurIPS, 2024, 37: 50772-50804.
> >
> > [2] Niu C, Qiao H, Chen C, et al. Zero-shot generalist graph anomaly detection with unified neighborhood prompts. IJCAI, 2025.
> >
> > [3] Xiong Zhang, Zhenli He, Changlong Fu, Cheng Xie. IA-GGAD: Zero-shot Generalist Graph Anomaly Detection via Invariant and Affinity Learning. IJCAI, 2025.
> >
> > [4] Qiao H, Niu C, Chen L, et al. AnomalyGFM: Graph foundation model for zero/few-shot anomaly detection. SIGKDD. 2025: 2326-2337.
> >
> > [5] Zhao Y, Liu Y, Li S, et al. Freegad: A training-free yet effective approach for graph anomaly detection. CIKM. 2025: 4379-4389.
> >
> > [6] Xu Y, Chen J, Peng Z, et al. Court of LLMs: Evidence-Augmented Generation via Multi-LLM Collaboration for Text-Attributed Graph Anomaly Detection. ACM MM. 2025: 2437-2446.
> >
> > **>6. Updating References from arXiv Preprints to Officially Published Versions**
> >
> > We sincerely appreciate your careful review of our manuscript and for pointing out the issues in the references. We apologize for the oversight in citing early arXiv versions for some papers that have already been formally published. This was indeed a lapse in our literature tracking and manuscript revision process. We fully agree with your point that citing the final, peer-reviewed, officially published versions is not only a matter of respecting the original authors' work but also a fundamental requirement for the rigor of academic writing.
> >
> > In response to your comments, we have conducted a thorough check and update of the entire reference list. For all works that have transitioned from arXiv preprints to formal publication, we have ensured that:
> >
> > (1) They are replaced with citations to the final conference or journal versions.
> >
> > (2) Complete official bibliographic information is provided, including venue name, publication year, volume/issue numbers, page ranges, or article IDs when applicable.

---

### Comment · Area_Chair_pKkf · 2025-11-28

Dear Reviewers,

Thank you for your valuable time and expertise in reviewing this paper.

The authors have now submitted their rebuttal. We would appreciate it if you could review their responses and assess whether your concerns have been addressed, if you haven't done this.

Best regards,

AC

---

### Author Response · Authors · 2025-12-01

Dear AC:

We hope this email can help reduce your workload. We sincerely appreciate your time and effort in reviewing our manuscript and our responses.

The reviewer Ad2j's main concerns were: 1) no comparison with naïve pseudo-labeling; 2) unclear role of graph structural properties; and 3) insufficient explanation of parameters, related work, and references. In response, we added experiments showing our method surpasses naïve pseudo-labeling. We also clarified that graph structure filters candidate anomalies, while labeled normal samples guide semantic detection. All other concerns were fully addressed in the revision.

The reviewer DW3F's main concerns were: 1) robustness with noisy graph structures 2）negligible weight of PIS in scoring function 3）sensitivity to pseudo-anomaly ratio selection
In response, we clarified TAQ-GAD's dual-driven framework integrating both topology and features. We also demonstrated PIS's tangible contribution through ablation studies. Also, we justified parameter robustness via comprehensive sensitivity analyses. Since the review comment thread was closed before our interaction with this reviewer was finished, we provide additional experimental explanations to address the reviewer’s remaining points of confusion and to demonstrate the necessity of our proposed PIS component; furthermore, through these newly added experiments, we also show convincingly that the reviewer’s concern cannot be regarded as a weakness of our method, as it stems solely from the choice of parameter settings.

The reviewer AXae raised four primary concerns regarding our approach: (1) methodological limitations, (2) label dependency, (3) potential structural bias in heterogeneous graphs, and (4) generalizability. In response, we systematically addressed each issue by clarifying that TAQ-GAD does not require true anomaly labels and does not rely on strong clustering assumptions. We further explained that the NBS and PIS components act as initial filters, while the TAE module enables refined learning beyond these heuristic rules. We also conducted all experiments under standardized low-label settings and demonstrated robust performance on heterogeneous graphs. The reviewer acknowledged that our responses largely resolved their concerns and indicated an intention to raise the score.

The reviewer 4ajr questioned: 1) the focus on topological anomalies, 2) hyperparameter selection, 3) label-flipping threshold determination, and 4) node selection for centroid computation. We clarified that TAQ-GAD integrates topology and features for comprehensive anomaly detection, elaborated hyperparameter settings and sensitivity analysis. We also clarified the label-flipping criterion, and detailed the use of high-confidence pseudo-labels in centroid updates.

Overall, due to unforeseen circumstances, we were unable to complete our interaction with some reviewers before the deadline. Fortunately, the remaining concerns mainly involved interpretational issues such as parameter choices and threshold settings. Our newly added experimental results and explanations have effectively resolved these concerns—for example, reviewer AXae has already decided to raise their score.

We would like to once again express our gratitude for your review and valuable guidance on this work.

Sincerely,

Paper 22854 authors

---

### Meta-Review · Area_Chair_6nif · 2026-01-07

**Summary:**

The reviewers converged on three points: (i) the pseudo-labeling mechanism needs stronger evidence that it outperforms naïve baselines; (ii) the topology-driven components (NBS/PIS) may be brittle under noisy or heterogeneous structure and their weights appear imbalanced; (iii) the label-scarce setup was not strictly aligned with prior work, raising fairness concerns. Authors supplied new ablations, sensitivity curves, and unified label-ratio experiments; one reviewer upgraded, while two maintained “borderline-below” ratings, still doubting robustness and component contribution.

**Reviewer Concerns:**

Addressed: comparison with random/low-degree pseudo labels; clarification of graph-structure role; extra baselines; corrected label-ratio table; PIS ablation showing small but consistent gain; reference updates.
Outstanding: residual skepticism that topology priors suffice on truly noisy or heterophilic graphs; negligible PIS weight still looks like “dead code”; no new runs on dynamic/multi-relational data; one reviewer explicitly kept score unchanged.

**Reviewer Scores:**

Ad2j: 4 → 4
DW3F: 4 → 4
AXae: 4 → 6 (stated intention to raise, system locked)
4ajr: 8 → 8

---

### Decision · Program_Chairs · 2026-01-26

Accept (Poster)